# A Rho signaling network links microtubules to PKD controlled carrier transport to focal adhesions

Stephan A Eisler[1†], Filipa Curado[2†], Gisela Link[2], Sarah Schulz[2], Melanie Noack[1], Maren Steinke[2], Monilola A Olayioye[1,2], Angelika Hausser[1,2]*

[1]Stuttgart Research Center Systems Biology, University of Stuttgart, Stuttgart, Germany; [2]Institute of Cell Biology and Immunology, University of Stuttgart, Stuttgart, Germany

**Abstract** Protein kinase D (PKD) is a family of serine/threonine kinases that is required for the structural integrity and function of the Golgi complex. Despite its importance in the regulation of Golgi function, the molecular mechanisms regulating PKD activity are still incompletely understood. Using the genetically encoded PKD activity reporter G-PKDrep we now uncover a Rho signaling network comprising GEF-H1, the RhoGAP DLC3, and the Rho effector PLCε that regulate the activation of PKD at trans-Golgi membranes. We further show that this molecular network coordinates the formation of TGN-derived Rab6-positive transport carriers delivering cargo for localized exocytosis at focal adhesions.

DOI: https://doi.org/10.7554/eLife.35907.001

*For correspondence:
angelika.hausser@izi.uni-stuttgart.de

†These authors contributed equally to this work

Competing interests: The authors declare that no competing interests exist.

## Introduction

The cytoskeleton controls cell morphology, polarity, division and movement but also organelle positioning, integrity and function. The Golgi apparatus is the central sorting and modification station of transmembrane and secretory proteins and has a perinuclear localization in interphase mammalian cells. The integrity and position of the Golgi complex are primarily governed by the microtubule network, with a smaller contribution of the actin cytoskeleton (*Allan et al., 2002*). Microtubules are also implicated in the transport of vesicles from the Golgi and localized exocytosis. Furthermore, during directed migration of cells, microtubule remodeling positions the Golgi complex towards the leading edge to support polarized transport and is further required for the rapid turnover of focal adhesions (FAs) (reviewed in [*Kaverina and Straube, 2011*]). The crosstalk between microtubules and the Golgi complex is coordinated by signaling factors including small GTPases such as Cdc42, Arf1, and Rho and their respective GEF and GAP proteins (reviewed in [*Millarte and Farhan, 2012*]). For example, Arf1-mediated recruitment of the Cdc42-specific GAP ARHGAP21 (ARHGAP10) to Golgi membranes controls Cdc42 activity and microtubule-dependent positioning of the Golgi complex (*Dubois et al., 2005*; *Hehnly et al., 2010*). Although Rho hyperactivation has been reported to lead to Golgi complex fragmentation (*Zilberman et al., 2011*), the physiological relevance of this observation is not clear. ARHGEF2/GEF-H1 is a RhoGEF that is sequestered by microtubules and it is further regulated downstream of G-protein coupled receptor (GPCR) ligands (*Meiri et al., 2009*, *2014*) and ERK signaling (*Guilluy et al., 2011*; *Kakiashvili et al., 2009*; *Scott et al., 2016*; *Waheed et al., 2010*). GEF-H1 has been involved in exocytosis at the plasma membrane, FA turnover, response to mechanical forces, cell migration, and cell polarity (reviewed in [*Pathak and Dermardirossian, 2013*]), however, so far it has not been connected to vesicle fission at the Golgi complex.

The protein kinase D (PKD) family of serine/threonine kinases consisting of PKD1, PKD2, and PKD3 in mammals is enriched at the trans-Golgi network (TGN) where it coordinates the fission of

vesicles destined for the plasma membrane (*Liljedahl et al., 2001*; *Malhotra and Campelo, 2011*). PKD is recruited to and activated at the TGN by the interaction with diacylglycerol (DAG) and Arf1 (*Baron and Malhotra, 2002*; *Pusapati et al., 2010*) and through direct phosphorylation by PKCη (*Díaz Añel and Malhotra, 2005*). Because the level of DAG in TGN membranes controls the localization and activity of PKD, the pathways contributing to the local production of DAG are critical regulators of PKD-dependent transport carrier formation. For example, PKD phosphorylates and activates the lipid kinase PI4KIIIβ thereby increasing PtdIns(4)P levels at the TGN (*Hausser et al., 2005*). This, in turn, recruits the ceramide transfer protein CERT from the ER to the Golgi membranes, where CERT contributes to PKD activation by providing ceramide as a precursor for further DAG production (*Fugmann et al., 2007*). Additionally, the family of PI-PLC enzymes, which generate DAG and inositol trisphosphate ($IP_3$) through the hydrolysis of $PtdIns(4,5)P_2$, are implicated in PKD activation downstream of Gβγ subunits (*Bard and Malhotra, 2006*; *Díaz Añel, 2007*). We previously reported that Golgi-localized PKD activity is enhanced by drug-induced microtubule depolymerization (*Fuchs et al., 2009*) but how the microtubule network signaled to the TGN remained elusive. Here, we identify GEF-H1 to mediate PKD activation at the TGN upon nocodazole-induced release from microtubules. We provide evidence that GEF-H1 signals through RhoA and PLCε to control steady state and also G-Protein coupled receptor (GPCR)-induced activity of PKD at the TGN membranes. Most importantly, we show that at the TGN, PKD activated downstream of Rho stimulates the fission of Rab6-positive exocytic carriers delivering cargo to FAs. Consequently, through GEF-H1, the microtubule network controls anterograde trafficking from the TGN.

## Results

### RhoA activates PKD at the TGN

Nocodazole is a strong activator of PKD at TGN membranes (*Fuchs et al., 2009*), suggesting that the microtubule network not only plays a structural role in maintaining Golgi complex position and shape but also signals to the Golgi complex to coordinate its function. Microtubule disruption was described to release the microtubule-associated RhoGEF GEF-H1 leading to the activation of the Rho GTPases RhoA and RhoB (*Arnette et al., 2016*; *Chang et al., 2008*). We therefore first analyzed whether RhoA can activate PKD at the TGN. To do so, we employed a Golgi-localized PKD activity reporter previously developed in our lab, termed G-PKDrep, which specifically detects PKD activity at the TGN (*Fuchs et al., 2009*). When cells expressed constitutive active RhoA Q63L, the reporter phosphorylation and thus PKD activity significantly increased up to twofold (0.2284 (median control)) versus 0.4826 (median RhoA Q63L)) whereas expression of wild type (wt) RhoA had no effect compared to the control cells (*Figure 1A and B*). As reported previously, expression of RhoA Q63L resulted in Golgi fragmentation (*Zilberman et al., 2011*) and stress fiber formation (*Figure 1A and C*). In support of a local role for RhoA in the activation of PKD we could detect GFP-tagged RhoA Q63L at TGN membranes indicated by the co-localization with the trans-Golgi marker p230 (*Figure 1C*). Similarly, active RhoB Q63L localized to the TGN and activated PKD at this site (*Figure 1—figure supplement 1A–C*). We further quantified the co-localization of RhoA and RhoB with Rab6, a TGN-localized Rab GTPase involved in the fission of transport carriers (*Miserey-Lenkei et al., 2010*), using the Manders' coefficient. Indeed, the co-localization of constitutive active RhoA and RhoB with Rab6 was significantly higher when compared to their wild-type counterparts, further supporting a role for active Rho at TGN membranes (*Figure 1—figure supplement 1D*).

### Rho-mediated PKD activation at the TGN requires GEF-H1

GEF-H1 directly binds to microtubules and, when bound, is inactive (*Krendel et al., 2002*). With increasing microtubule depolymerization, GEF-H1 is released and free to activate Rho (*Arnette et al., 2016*; *Chang et al., 2008*). To examine whether GEF-H1 is responsible for mediating PKD activation downstream of RhoA, we knocked down GEF-H1 using two independent siRNAs. Indeed, in GEF-H1-depleted cells nocodazole-induced PKD activation was strongly reduced (*Figure 2A*, *Figure 2—figure supplement 1*). GEF-H1 can be activated downstream of ERK signaling (*Fujishiro et al., 2008*; *Guilluy et al., 2011*; *Kakiashvili et al., 2009*; *Waheed et al., 2010*). In HEK293T cells, nocodazole treatment strongly induced the MAPK pathway. Simultaneous MEK1/2 inhibition, however, did not impair phosphorylation of PKD (*Figure 2—figure supplement 2A*)

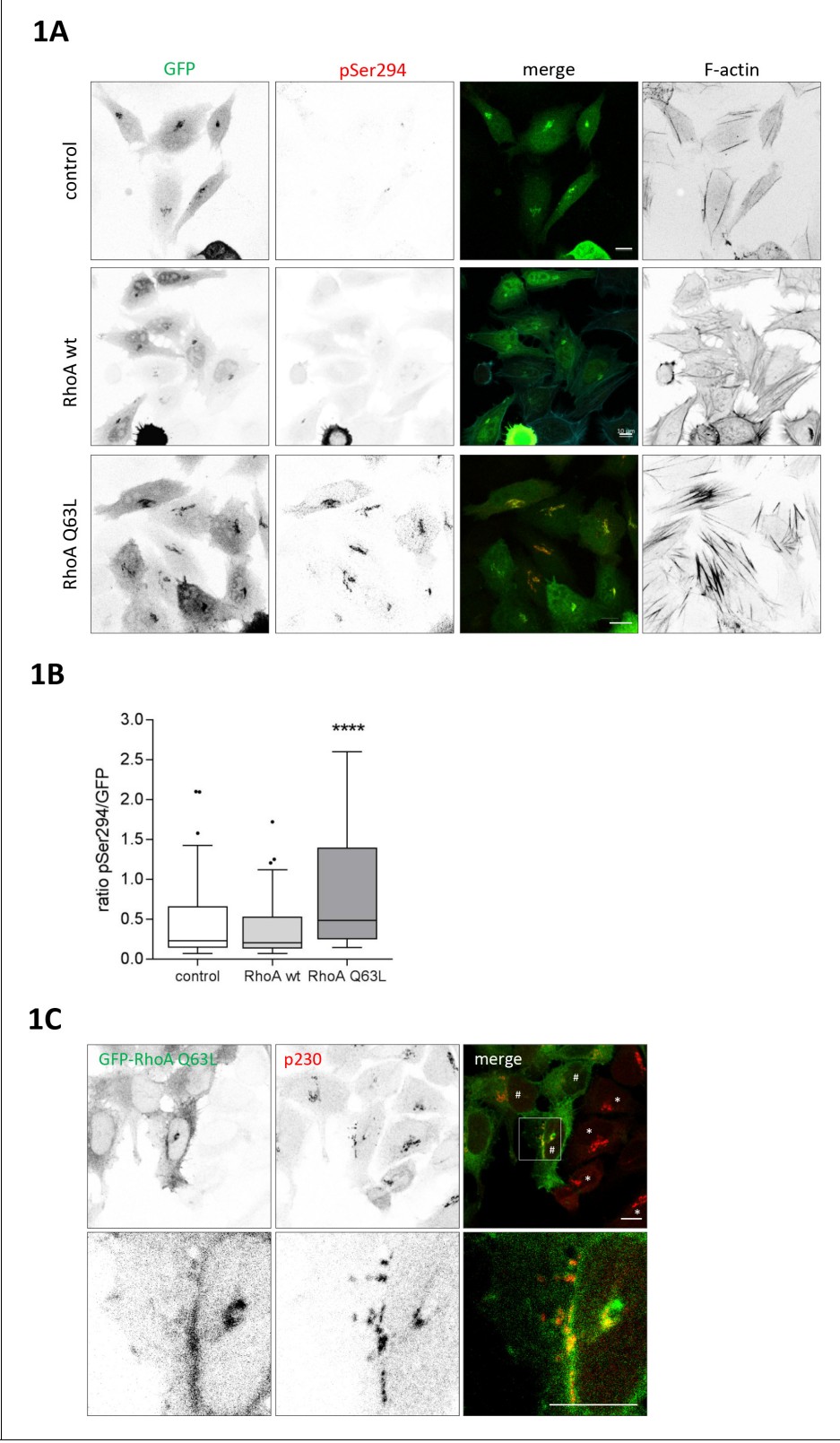

**Figure 1.** RhoA activates PKD at the TGN. HeLa cells were transfected with G-PKDrep plus control vector, or a plasmid encoding HA-tagged RhoA wt or RhoA Q63L. One day after transfection cells were fixed and stained for G-PKDrep phosphorylation (pSer294) and F-actin (Alexa633-labelled phalloidin). Images were quantified by ratiometric calculation of GFP and Alexa546-labelled pSer294 signal. (**A**) Shown are representative confocal images, scale bar 10 μm. (**B**) The box plot shows the results of three independent experiments. Center lines show the medians; box limits indicate the

*Figure 1 continued on next page*

*Figure 1 continued*

25th and 75th percentiles as determined by GraphPad Prism 7 software; whiskers extend 1.5 times the interquartile range from the 25th and 75th percentiles, outliers are represented by dots. n = 72 sample points each. The significance of differences was analyzed by a one-way ANOVA (Kruskal-Wallis test) followed by a Dunn's multiple comparison test. ****$p<0.0001$ (control vs. RhoA Q63L). (C) HeLa cells were transfected with a plasmid encoding GFP-tagged RhoA Q63L, fixed 24 hr later, and stained for the trans Golgi protein p230 (red). Shown is a confocal image, scale bar 10 μm. Yellow regions indicate co-localization. * indicates cells with a compact Golgi complex, # indicates cells with a fragmented Golgi complex.
DOI: https://doi.org/10.7554/eLife.35907.002
The following figure supplement is available for figure 1:

**Figure supplement 1.** RhoB activates PKD at the TGN.
DOI: https://doi.org/10.7554/eLife.35907.003

proving that GEF-H1-mediated PKD activation is independent of ERK. In agreement with our GEF-H1 depletion results, expression of GEF-H1 wt and GEF-H1 C53R, a mutant that does not bind microtubules and is thus constitutively active (*Krendel et al., 2002*), was sufficient to activate PKD (*Figure 2B*). Of note, GEF-H1 C53R induced PKD activation was stronger compared to GEF-H1 wt. Analogous to the expression of active RhoA, expression of GEF-H1 C53R induced fragmentation of the Golgi complex whereas in cells expressing GEF-H1 wt the Golgi complex remained compact (*Figure 2—figure supplement 2B*). To corroborate the GEF-H1-induced PKD activation at the TGN we again used G-PKDrep. In cells expressing GEF-H1 wt, the reporter phosphorylation was significantly enhanced compared to the control cells without ectopic GEF-H1 expression (*Figure 2C*). In agreement with the quantitative Western Blot results (*Figure 2B*), reporter phosphorylation was stronger in the case of GEF-H1 C53R expression (*Figure 2C*).

We next sought to prove that the RhoA pool activated by GEF-H1 is present on TGN membranes by tracking RhoA activity in living cells using an unimolecular RhoA activity FRET biosensor (*Pertz et al., 2006*). This RhoA biosensor was described to co-localize with Golgi membrane markers (*Pertz et al., 2006*). In HeLa cells the RhoA biosensor also displayed a partial co-localization with a co-expressed Golgi marker protein (mRuby-Golgi-7) (*Figure 2—figure supplement 3*). We stimulated control and GEF-H1 depleted cells with nocodazole and measured RhoA activity over time by ratiometric FRET imaging (see *Figure 2D* for FRET images at 0 and 30 min). As previously reported (*Reinhard et al., 2016*), nocodazole stimulation led to a GEF-H1-dependent increase in RhoA activity in the cell periphery (see images in *Figure 2D*). At the Golgi, RhoA activity increased steadily and reached a plateau 10 min after nocodazole treatment in control cells. In GEF-H1 depleted cells, however, no increase in RhoA activity at the cell periphery and the Golgi was observed (see images and graph in *Figure 2D*). Our data thus provide strong support for GEF-H1 promoting increased Rho activity in the cell periphery and at the Golgi membranes.

## The Rho effector PLCε is required for PKD activation at the TGN

Rho GTPases engage various downstream effector proteins, some of which were previously linked to Golgi complex function. For example, mDia1 was reported to mediate RhoA-induced Golgi fragmentation (*Zilberman et al., 2011*), however, in our experimental system, depletion of mDia1 did not affect RhoA-mediated PKD activation at Golgi membranes (*Figure 3A, B and C*). The Rho effector kinases ROCK1 and 2 were shown to mediate cell contractility downstream of nocodazole-induced Rho activation (*Chang et al., 2008*) and expression of constitutive active ROCK1 induced Golgi fragmentation (*Orlando and Pittman, 2006*), but neither the single, nor the combined depletion of ROCK1 and ROCK2 affected Golgi-associated PKD activity (*Figure 3—figure supplement 1A*). Pharmacological ROCK inhibition by H1152 also failed to block RhoA Q63L-induced PKD activation (*Figure 3—figure supplement 1B*).

PLCε is a Rho effector protein found to localize to the Golgi complex in cardiac myocytes (*Zhang et al., 2013*) where it hydrolyzes PtdIns(4)P resulting in the production of DAG. We thus investigated whether PLCε was involved in PKD activation. Indeed, depletion of PLCε using independent siRNAs significantly reduced RhoA-mediated PKD activation at the TGN (*Figures 3A and B and C* and *Figure 3—figure supplement 2*). Furthermore, in cells depleted of PLCε, nocodazole treatment no longer resulted in the PKD activation seen in the control cells (*Figure 3D* and *Figure 2—figure supplement 1*), confirming that PLCε is downstream of RhoA in PKD activation at Golgi membranes. To address whether the GEF-H1 dependent PKD activation requires PLCε, we

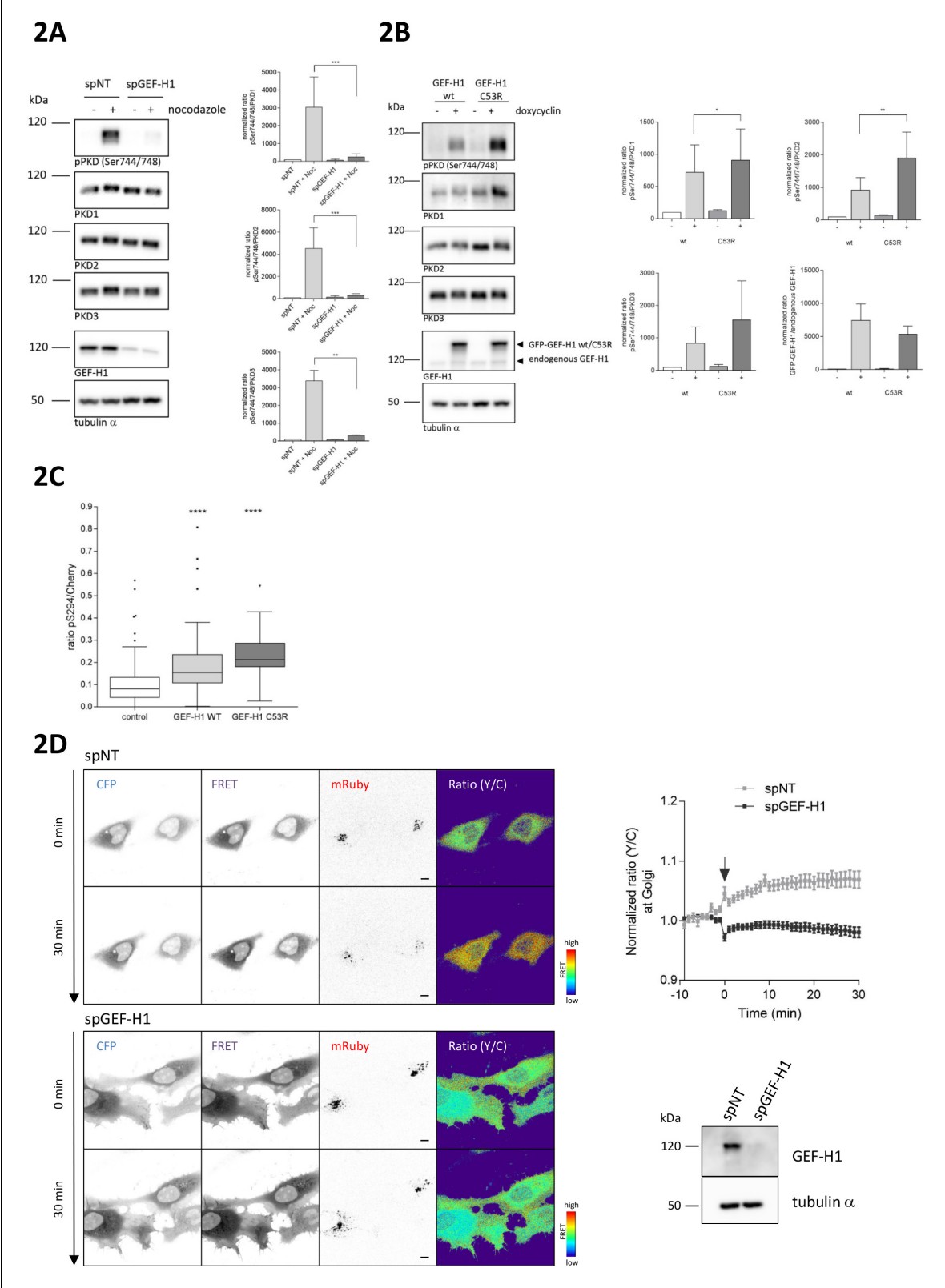

**Figure 2.** Nocodazole-mediated PKD activation at the TGN requires GEF-H1. (**A**) HEK293T cells were transfected with spGEF-H1, spNT was used as a control. Three days after transfection cells were stimulated with nocodazole for 60 min, lysed and analyzed for PKD activity (pPKD (Ser744/748) and expression of PKD1, 2, and 3. Detection of alpha tubulin served as a loading control. Successful depletion of GEF-H1 was verified by detection with a GEF-H1 specific antibody. Shown is a representative Western blot. The integrated density of the pPKD and PKD1-3 signal was measured, corrected for

*Figure 2 continued on next page*

*Figure 2 continued*

background signals and the ratio (pPKD/PKD) calculated. Data were normalized to the control (spNT -nocodazole), which was set to 100. The graphs show the mean ± SEM of three independent experiments. The significance of differences was analyzed by a Ratio paired t-test (two-tailed). ***p=0.0007 (ratio pPKD Ser744/748/PKD1) and p=0.0002 (ratio pPKD Ser744/748/PKD2), **p=0.0014 (ratio pPKD Ser744/748/PKD3). (B) FlpIn T-Rex 293 EGFP-GEF-H1 wt or FlpIn T-REx 293 EGFP-GEF-H1 C53R cells were left untreated (-) or treated with doxycyline for 18 hr (+). Cells were lysed and analysed for PKD activity, PKD1-3 expression and endogenous as well as ectopic GEF-H1 expression. Detection of alpha tubulin served as a loading control. Shown is a representative Western blot. Quantification of Western blot data was done as described in *Figure 2A*. Data were normalized to the control (GEF-H1 wt untreated), which was set to 100. The graphs show the mean ± SEM of three independent experiments. The significance of differences was analyzed by a Ratio paired t-test (two-tailed). *p=0.0365 (ratio pPKD Ser744/748/PKD1) and **p=0.0023 (ratio pPKD Ser744/748/PKD2). (C) FlpIn T-Rex HeLa EGFP-GEF-H1 wt or EGFP-GEF-H1 C53R cells were transfected with a Cherry-tagged G-PKDrep. One day after transfection cells expression of GEF-H1-GFP was induced by doxycyline. One day later, cells were fixed and stained for G-PKDrep phosphorylation (pSer294). Images were quantified by ratiometric calculation of Cherry and Alexa633-labelled pSer294 signal. Cells without doxycycline treatment were used as control. The box plot shows the results of three independent experiments. Center lines show the medians; box limits indicate the 25th and 75th percentiles as determined by GraphPad Prism 7 software; whiskers extend 1.5 times the interquartile range from the 25th and 75th percentiles, outliers are represented by dots. n = 89, 56, 23 sample points. The significance of differences was analyzed by a one-way ANOVA (Kruskal-Wallis test) followed by a Dunns multiple comparison test. ****p<0.0001 (control vs GEF-H1 wt or GEF-H1 C53R). (D) HeLa cells were transfected with the indicated siRNAs. Two days later, cells were transfected with the RhoA Biosensor along with mRuby-Golgi-7 (ratio 6:1) and, after 24 hr, stimulated with nocodazole and imaged. Left panel, representative images taken at 0 and 30 min of stimulation are shown; scale bar 10 µm. Right panel, the graph shows the mean ± SEM of at least 40 cells imaged in two independent experiments. The arrow indicates the time point of nocodazole stimulation. Successful depletion of GEF-H1 was verified by Western Blot analysis. Detection of alpha tubulin served as a loading control.

DOI: https://doi.org/10.7554/eLife.35907.004

The following figure supplements are available for figure 2:

**Figure supplement 1.** Validation of the requirement of PLCε and GEF-H1 for nocodazole-induced PKD activation.
DOI: https://doi.org/10.7554/eLife.35907.005

**Figure supplement 2.** GEF-H1 mediated PKD activation is independent of ERK
DOI: https://doi.org/10.7554/eLife.35907.006

**Figure supplement 3.** The RhoA biosensor localizes to the Golgi complex.
DOI: https://doi.org/10.7554/eLife.35907.007

expressed GEF-H1 C53R in cells depleted of PLCε and monitored PKD activity by Western blot analysis. In control cells, GEF-H1 C53R potently activated PKD whereas the loss of PLCε diminished the GEF-H1 C53R-mediated PKD activation (*Figure 3E*).

Because PLCε is responsible for the production of DAG, we reasoned that PLCε should also be involved in the maintenance of basal PKD activity at Golgi membranes. To test this assumption, we quantitatively analyzed the subcellular localization of a kinase-dead PKD1-GFP fusion protein commonly used as a sensor for DAG (*Baron and Malhotra, 2002*) by measuring its fluorescence intensity at the Golgi complex and in the cytoplasm. As shown in *Figure 4A* (top), PKD1kd was enriched at Golgi membranes and a smaller portion was also present in the cytosol. Quantification revealed that in cells depleted of PLCε, the amount of PKD1kd at Golgi membranes was reduced compared to control cells (*Figure 4A*, bottom). This suggests that the loss of PLCε decreases DAG at and thus PKD localization to Golgi membranes. Consequently, basal PKD activity should be dependent on PLCε as well. Indeed, quantitative measurement of G-PKDrep phosphorylation revealed that PLCε depletion led to a modest but significant reduction in PKD activity at TGN membranes compared to control cells (*Figure 4B and C*). Thus, PLCε is involved in the regulation of both, basal PKD activity and nocodazole-induced PKD activation at Golgi membranes.

GPCRs, in particular the family of protease activated receptors (PAR), which are activated by trypsin, thrombin and factor Xa, as well as LPA responsive receptors trigger the activation of GEF-H1, independently of microtubule depolymerization (*Meiri et al., 2014*). Notably, in astrocytes, thrombin activates PKD downstream of PLCε (*Dusaban et al., 2013*). Moreover, activation of PAR2 through trypsin induced PKD activity at the Golgi complex (*Jensen et al., 2016*), raising the question whether PAR signaling has a physiological function in connecting the plasma membrane with the Golgi complex through GEF-H1. Indeed, short trypsin stimulation strongly activated PKD as shown by activation loop phosphorylation of the kinase (*Figure 4D*). To address whether PARs are upstream of GEF-H1 and PLCε we stimulated HeLa cells expressing G-PKDrep with trypsin and measured PKD activation at the Golgi by ratiometric imaging in control and GEF-H1- and PLCε-depleted cells. Knockdown of PKD2 and PKD3 served as a control. Our results clearly show that trypsin

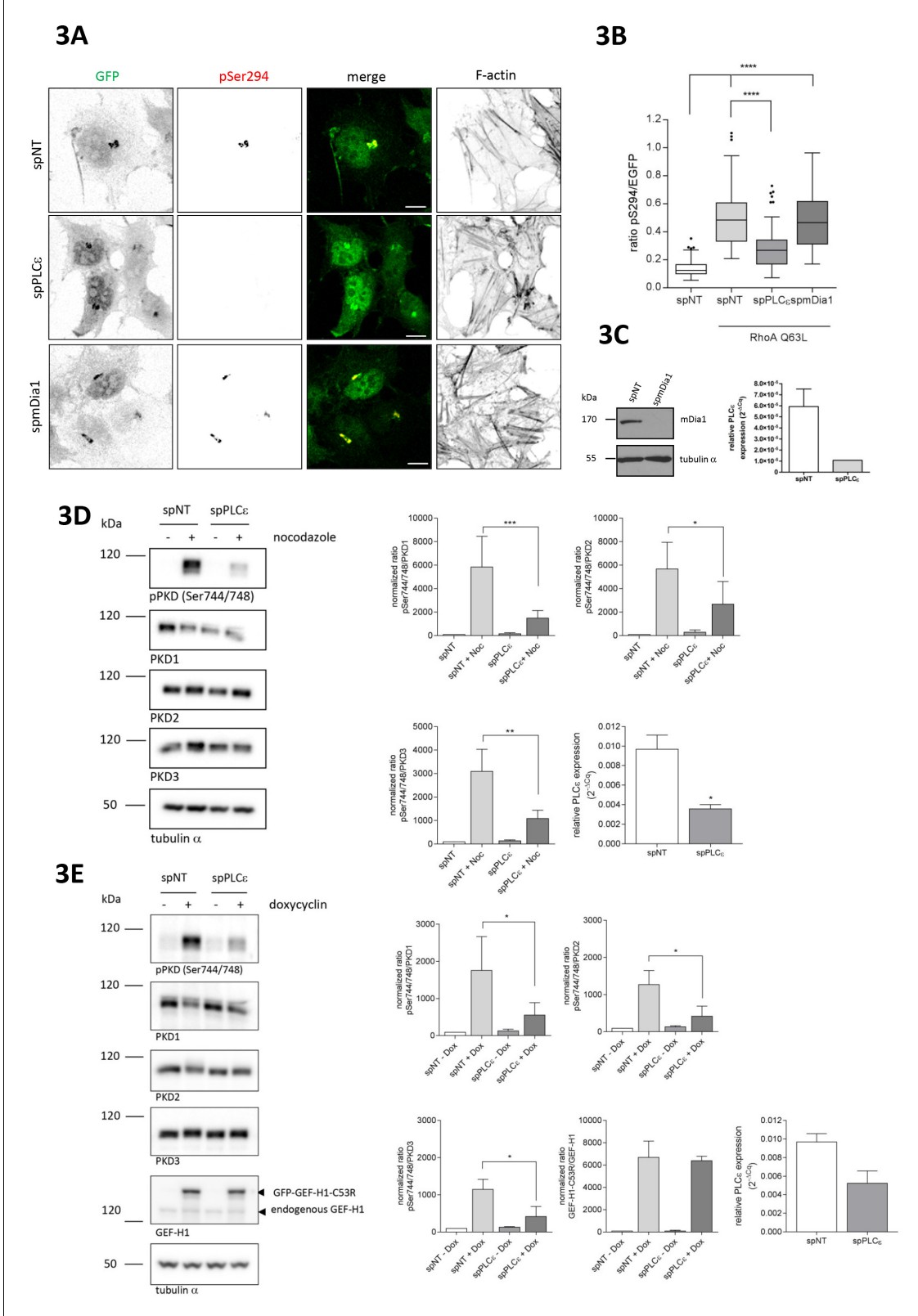

**Figure 3.** The Rho effector PLCε is required for GEF-H1 mediated PKD activation at the TGN. (**A**) HeLa cells were transfected with spRNAs as indicated, spNT was used as a control. Two days after transfection cells were transfected with G-PKDrep and RhoA Q63L and, after 24 hr, fixed, stained and analyzed as described in *Figure 1A*. Shown are representative confocal images, scale bar 10 μm. (**B**) The box plot shows the results of three independent experiments. Center lines show the medians; box limits indicate the 25th and 75th percentiles as determined by GraphPad Prism 7

*Figure 3 continued on next page*

*Figure 3 continued*

software; whiskers extend 1.5 times the interquartile range from the 25th and 75th percentiles, outliers are represented by dots. n = 90 sample points each. The significance of differences was analyzed by a one-way ANOVA followed by a Dunns multiple comparison test. \*\*\*p<0.0001. (C) Left panel, silencing efficiency of mDia1 was analyzed in lysates by immunoblotting using a mDia1-specific antibody. Equal loading was verified by detection of alpha tubulin. Right panel, successful depletion of PLCε was verified by RT-qPCR. Relative expression was calculated by normalization to GAPDH using the ΔCq method. Shown is the mean ± SEM of two independent experiments. (D) Left panel, HEK293T cells were transfected with the spPLCε, spNT was used as a control. Three days post transfection cells were left untreated or stimulated with nocodazole for 60 min. Detection of active PKD and expression of PKD1-3 was performed in cell lysates using specific antibodies. Equal loading was verified by detection of alpha tubulin. Shown is a representative Western blot. Right panel, quantification of Western blot data was done as described in *Figure 2A*. Data were normalized to the control (untreated spNT cells), which was set to 100. The graphs show the mean ± SEM of three independent experiments. The significance of differences was analyzed by a Ratio paired t-test (two-tailed). \*\*\*p=0.0007 (ratio pPKD Ser744/748/PKD1), \*p=0.0334 (ratio pPKD Ser744/748/PKD2), and \*\*p=0.0026 (ratio pPKD Ser744/748/PKD3). Successful depletion of PLCε was verified by RT-qPCR. Relative expression was calculated by normalization to Actin using the ΔCq method. Shown is the mean ± SEM of three independent experiments. The significance of differences was analyzed by a Ratio paired t-test (two-tailed). \*p=0.0157 (E) Left panel, FlpIn T-REx 293 EGFP-GEF-H1 C53R cells were transfected with spPLCε, spNT was used as a control. Two days later, GFP-GEF-H1 C53R expression was induced by doxycycline treatment. After one day, cells were lysed and analysed for PKD activity, PKD1-3 expression and endogenous as well as ectopic GEF-H1 expression. Detection of alpha tubulin served as a loading control. Shown is a representative Western blot. Right panel, quantification of Western blot data was done as described in *Figure 2A*. Data were normalized to the control (spNT cells without doxycycline), which was set to 100. The graphs show the mean ± SEM of three independent experiments. The significance of differences was analyzed by a Ratio paired t-test (two-tailed). \*p=0.0266 (ratio pPKD Ser744/748/PKD1), p=0.0106 (ratio pPKD Ser744/748/PKD2), and p=0.0152 (ratio pPKD SerS744/748/PKD3). Successful depletion of PLCε was verified by RT-qPCR. Relative expression was calculated by normalization to Actin using the ΔCq method. Shown is the mean ± SEM of three independent experiments.

DOI: https://doi.org/10.7554/eLife.35907.008

The following figure supplements are available for figure 3:

**Figure supplement 1.** ROCK1 and ROCK2 are not involved in PKD activation at the TGN.
DOI: https://doi.org/10.7554/eLife.35907.009

**Figure supplement 2.** Validation of PLCε as Rho-effector protein in PKD activation at the TGN using an independent siRNA.
DOI: https://doi.org/10.7554/eLife.35907.010

treatment significantly activated PKD at the Golgi complex in control cells (*Figure 4E and F*). However, in cells depleted of GEF-H1, PLCε or PKD2/3, trypsin-mediated PKD activation was diminished compared to control cells (*Figure 4F*). Furthermore, short thrombin stimulation also led to strong phosphorylation of G-PKDrep which was fully blocked by treatment with the pan-PKD inhibitor CRT0066101 (*Figure 4—figure supplement 1*) placing PARs upstream of a GEF-H1-PKD signaling pathway.

## The RhoGAP DLC3 counterbalances GEF-H1-mediated PKD activation at the TGN

Rho GTPase activation requires the spatially coordinated action of specific GEFs and GAPs. Recently, we identified Deleted in Liver Cancer 3 (DLC3), a RhoGAP protein with specificity for RhoA (*Holeiter et al., 2012*) to control RhoA-GTP levels at Golgi membranes. Depletion of DLC3 in HeLa cells significantly increased basal RhoA activity at this compartment resulting in fragmentation of the Golgi complex (*Braun et al., 2015*). To address whether DLC3 could be a counterplayer of GEF-H1 in PKD activation at the Golgi complex, we depleted DLC3 in HeLa cells and analyzed PKD activity at the TGN using the G-PKDrep phosphorylation as a read out (*Figure 5A*). Importantly, we detected a significant increase in PKD activity when DLC3 was absent (*Figure 5B*). To corroborate this finding, we performed a set of rescue experiments. Along with DLC3, we depleted either the Rho effector PLCε or the RhoGEF GEF-H1. If DLC3 and GEF-H1 control the same RhoA pool, depletion of GEF-H1 or the Rho effector PLCε should rescue the PKD activation induced by DLC3 depletion. Indeed, when both proteins, DLC3 plus GEF-H1 or DLC3 plus PLCε were depleted, the significant increase in PKD activity as observed in DLC3-depleted cells was fully rescued (*Figure 5C and D*). Our results thus indicate that GEF-H1, Rho and PLCε are in a linear signaling pathway that links the microtubule network with PKD activity at the TGN and is counterbalanced by DLC3.

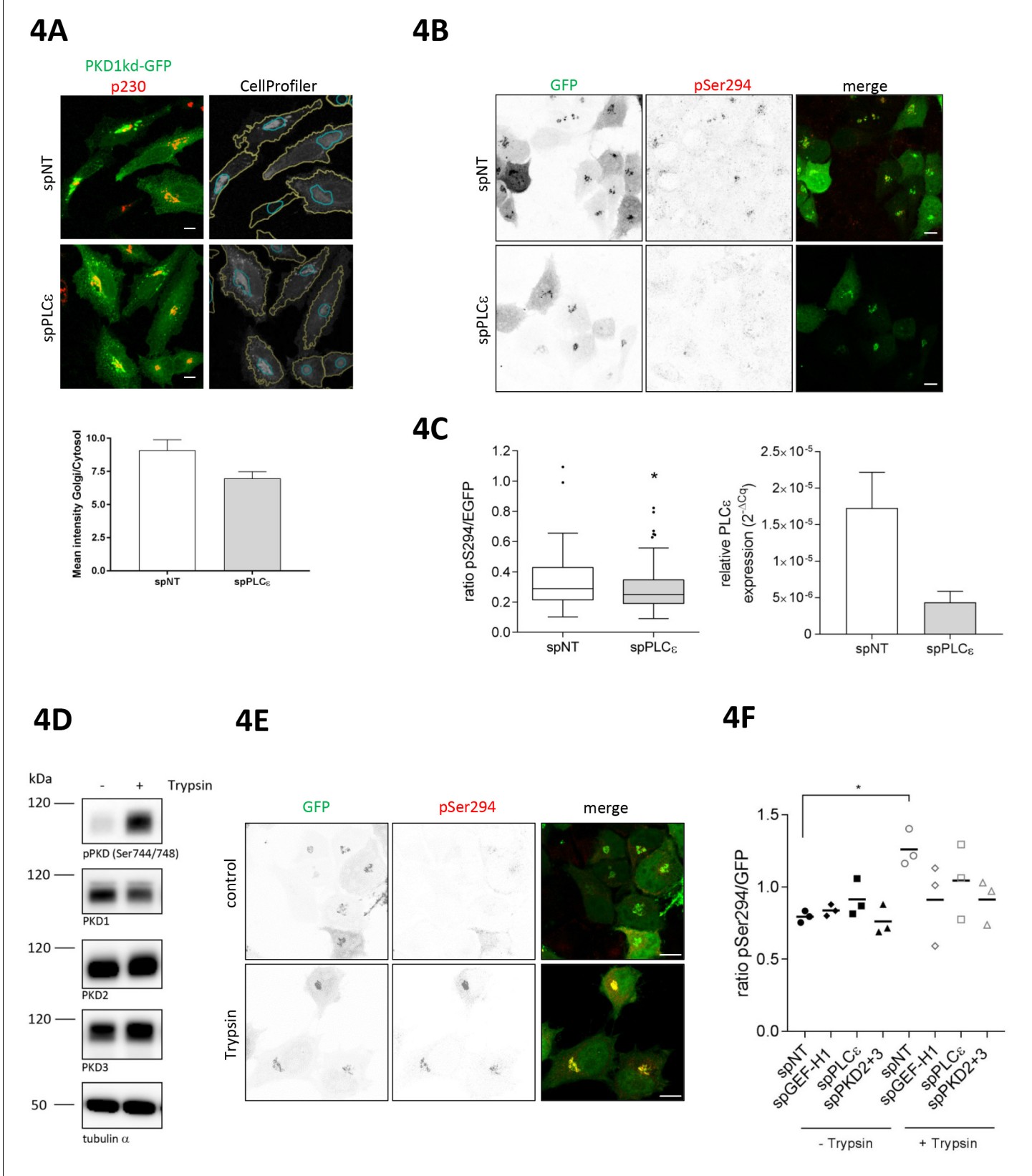

**Figure 4.** PLCε is required for basal and trypsin-induced PKD activity at the TGN. (**A**) Top panel, HeLa cells were transfected with spPLCε, spNT was used as a control. Two days later, cells were transfected with a plasmid encoding PKD1kd-GFP. After 24 hr, cells were fixed and stained for p230. *Figure 4 continued on next page*

*Figure 4 continued*

Representative confocal images are shown; scale bar 10 µm. CellProfiler images show the masks defining the Golgi complex (blue) and the cytosol (yellow) merged with the PKD1kd-GFP image (shown in greyscale). Bottom panel, the fluorescence intensities of the GFP signal were measured under both masks and the Golgi/Cytosol ratio was calculated for each cell. The graph shows the mean ± SEM of at least 150 cells analysed. (B) HeLa cells were transfected with siRNAs as described in E. Two days later, cells were transfected with G-PKDrep. Visualization and analysis of G-PKDrep phosphorylation was performed as described in *Figure 1A*. Shown are representative confocal images, scale bar 10 µm. (C) Left panel, the box plot shows the results of three independent experiments. Center lines show the medians; box limits indicate the 25th and 75th percentiles as determined by GraphPad Prism 7 software; whiskers extend 1.5 times the interquartile range from the 25th and 75th percentiles, outliers are represented by dots. n = 114, 104 sample points. The significance of differences was analyzed by a two-tailed t-test (Mann-Whitney test), *p<0.05. Right panel, successful depletion of PLCε was verified by RT-qPCR. Relative expression was calculated by normalization to GAPDH using the ΔCq method. Shown is the mean ± SEM of two independent experiments. (D) HEK293T cells were left untreated or stimulated with trypsin. Cells were lysed and detection of active PKD (pPKD Ser744/748) and expression of PKD1-3 was performed using specific antibodies. Equal loading was verified by detection of alpha tubulin. (E, F) HeLa cells were transfected with spRNAs as indicated, spNT was used as a control. Two days after transfection cells were transfected with G-PKDrep. 24 hr later, cells were left untreated or stimulated with trypsin, fixed, stained and analyzed as described in *Figure 1A*. (E) Shown are representative confocal images of the spNT control, scale bar 10 µm. (F) The scatter dot blot shows the result of three independent experiments, line indicates the mean. Each dot represents one experiment with at least 30 cells analysed. The significance of differences was analyzed by a one-way ANOVA (Friedman test) followed by a Dunn's multiple comparison test. *p=0.0306 (spNT – trypsin vs. spNT + trypsin). All other comparisons were not significant. Successful depletion of the proteins was verified by Western blot or RT-qPCR (data not shown).
DOI: https://doi.org/10.7554/eLife.35907.011

The following figure supplement is available for figure 4:

**Figure supplement 1.** PAR stimulation through thrombin and trypsin activates PKD at the Golgi.
DOI: https://doi.org/10.7554/eLife.35907.012

## GEF-H1, PLCε and PKD are required for localized delivery of Rab6 to FAs

At the TGN, PKD controls the fission of vesicles containing basolateral cargo destined for the plasma membrane. The carriers generated by PKD were named CARTS (CARriers of the TGN to the cell Surface) and are characterized by the presence of the small GTPases Rab6a and Rab8a (*Wakana et al., 2012*). A previous report showed that nocodazole treatment or expression of active RhoA recruits Rab8 to Golgi membranes (*Hattula et al., 2006*), prompting us to investigate whether this is regulated by PKD. We thus stained HeLa cells, which were transiently transfected with a plasmid encoding wt PKD1-GFP, for Rab8 or Rab6 and the trans-Golgi marker p230. In HeLa cells expressing PKD1-GFP, Rab8 co-localized with p230 and PKD1-GFP at trans-Golgi membranes whereas in cells without ectopic PKD1-GFP expression Rab8 was found on tubular and/or vesicular structures, which did not show an overlap with the trans-Golgi marker p230 (*Figure 6A*, top panel, histograms). Rab6 was strongly enriched at the TGN, indicated by the co-localization with p230 and PKD1-GFP, however, the localization of Rab6 at trans-Golgi membranes was independent of PKD1-GFP expression (*Figure 6A*, bottom panel, and histogram). Our observations were confirmed by a quantitative object-based co-localization analysis using CellProfiler software (*Figure 6—figure supplement 1*), which clearly showed that ectopic expression of PKD1-GFP or PKD2-GFP significantly increased the Golgi localization of Rab8, whereas in GFP vector transfected cells Rab8 localization remained unchanged (*Figure 6B*). Our results thus position PKD upstream of Rab8 recruitment to Golgi membranes.

Recently, Rab6 was connected to a FA-directed exocytosis pathway. The docking and fusion of Rab6 vesicles within the plasma membrane occurred preferentially near FAs (*Stehbens et al., 2014*). Moreover, Rab6-positive vesicles exit the Golgi at fission hotspots (*Miserey-Lenkei et al., 2017*). We thus speculated that the Rho-PKD signaling pathway might control the exit of Rab6 vesicles from the TGN and thus the localized delivery of these vesicles to FAs. To address this we performed TIRF analysis of HeLa cells expressing GFP-tagged Rab6A and DsRed2-tagged paxillin. We monitored the arrival of Rab6A vesicles in the vicinity of DsRed2-paxillin in cells transfected with a control siRNA (spNT), and in cells depleted of GEF-H1, PLCε, and PKD2 together with PKD3, respectively. As previously reported, in control cells Rab6A vesicles arrived at membrane domains proximal to FAs, stalled for a short period of time and then rapidly disappeared. Loss of GEF-H1, PLCε, or PKD2/PKD3, however, significantly decreased the number of Rab6A-positive vesicles arriving at FAs, with cells depleted of the two PKD isoforms showing the strongest effect (*Figure 7A and C*, *Figure 7—Video*

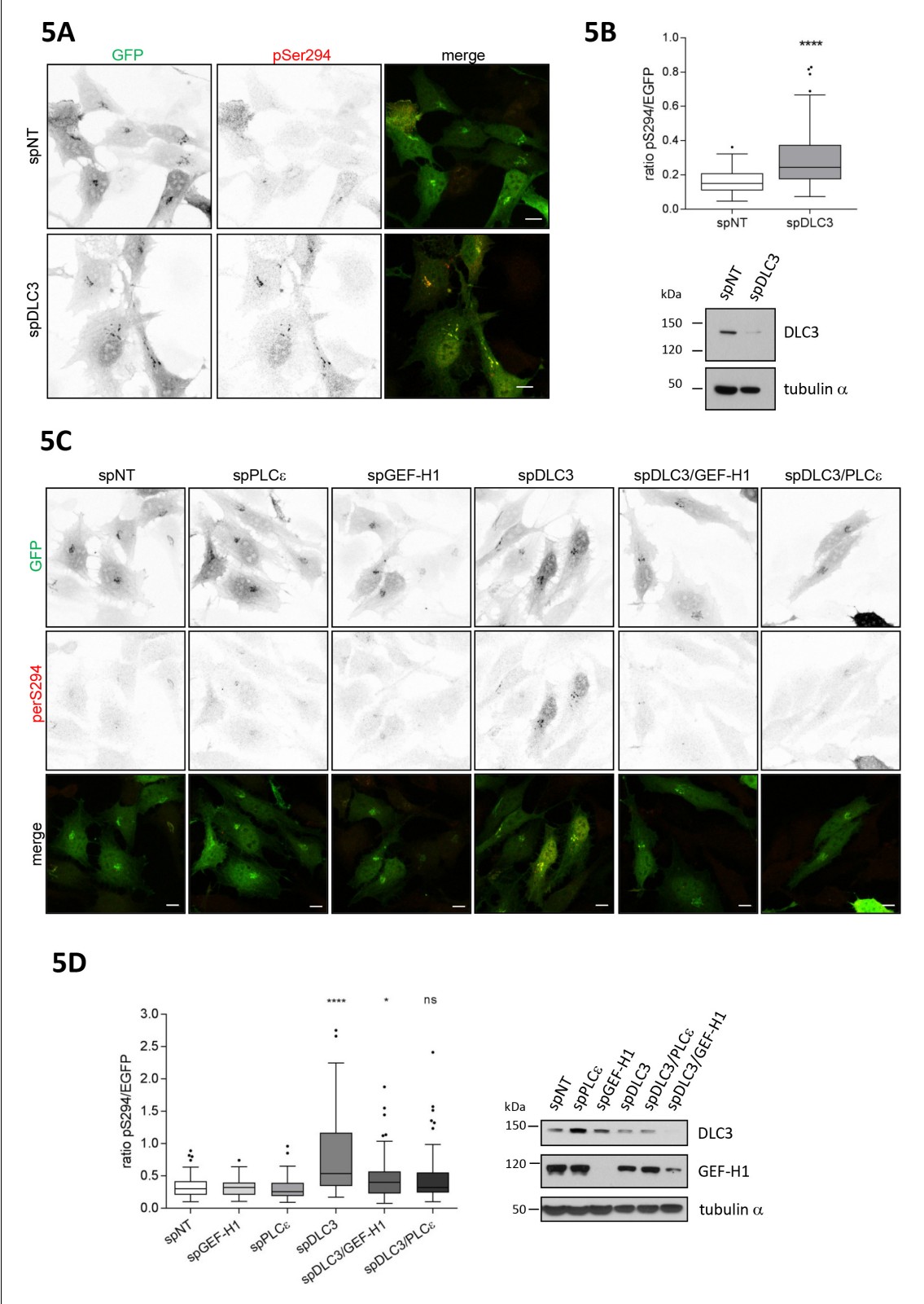

**Figure 5.** Loss of the RhoGAP DLC3 promotes PKD activation at the TGN. (**A**) HeLa cells were transfected with spRNAs as indicated. Two days after transfection cells were transfected with G-PKDrep and, after 24 hr, fixed, stained and analyzed by ratiometric measurement as described in *Figure 1A*. Representative confocal images are shown, scale bar 10 μm. (**B**) Top panel, the box plot shows the results of three independent experiments. Center lines show the medians; box limits indicate the 25th and 75th percentiles as determined by GraphPad Prism 7 software; whiskers extend 1.5 times the

*Figure 5 continued on next page*

*Figure 5 continued*
interquartile range from the 25th and 75th percentiles, outliers are represented by dots. n = 90 sample points each. The significance of differences was analyzed by a two-tailed t-test (Mann-Whitney test). ****p<0.0001. Bottom panel, silencing efficiency of DLC3 was analyzed in lysates by immunoblotting using a DLC3-specific antibody. Equal loading was verified by detection of alpha tubulin. (C) Cells were transfected with spRNAs as indicated. Two days after transfection cells were transfected with G-PKDrep and, after 24 hr, fixed, stained and analyzed by ratiometric imaging as described in 1A. Representative confocal images are shown, scale bar 10 μm. (D) The box plot shows the results of three independent experiments. Center lines show the medians; box limits indicate the 25th and 75th percentiles as determined by GraphPad Prism 7 software; whiskers extend 1.5 times the interquartile range from the 25th and 75th percentiles, outliers are represented by dots. n = 83, 73, 73, 75, 73, 75 sample points. The significance of differences was calculated by a one-way ANOVA (Kruskal-Wallis test) followed by a Dunn's multiple comparison test. ****p<0.0001, *p=0.0393. ns, not significant. Right panel, silencing efficiency of GEF-H1 and DLC3 were verified in lysates by immunoblotting using specific antibodies. Detection of alpha tubulin served as a loading control.
DOI: https://doi.org/10.7554/eLife.35907.013

1). Of note, the total number of FAs per cell was not affected by the loss of any of the proteins (*Figure 7B*). Interestingly, the depletion of GEF-H1 resulted in an upregulation of PLCε mRNA levels supporting the molecular connection of the two proteins (*Figure 7D*). Whether protein levels are also enhanced remains unknown because we could not detect PLCε by Western blot analysis. Notably, depletion of DLC3 slightly decreased the number of vesicles arriving at FAs compared to control cells although PKD activity was increased under these conditions (data not shown). However, as DLC3 depletion induced the fragmentation of the Golgi complex (see *Figure 5A* and [*Braun et al., 2015*]) it is likely that the polarized delivery of Rab6 vesicles to FAs is negatively affected. This also supports the hypothesis that Rho levels at the Golgi complex have to be tightly regulated to ensure proper vesicular trafficking. To connect GPCR signaling with the FA-directed exocytotic pathway, we stimulated control and PKD2/3 depleted HeLa cells with thrombin and quantified the Rab6 vesicles arriving at paxillin-positive FA over time by TIRF kymography. Our data clearly show that thrombin stimulation significantly increased the number of Rab6 vesicles arriving at FAs in control but not in PKD2/3 depleted cells (*Figure 7E*). We thus demonstrate that activation of PARs induces PKD-dependent localized exocytosis. We also followed the dynamic movement of CLIP170, a microtubule plus end tip, in HeLa cells expressing DsRed2-paxillin. As shown by the kymographs, in PLCε-depleted cells microtubule plus ends contacted FAs as efficiently as in control cells, proving that the loss of the Rho effector PLCε does not interfere with the microtubule-FA interaction per se (*Figure 7—figure supplement 1*). We thus conclude that perturbations of the GEF-H1/Rho/PLCε/PKD signaling axis affect Rab6 vesicles already at the level of the TGN.

To provide further proof that the GEF-H1/Rho/PLCε signaling axis regulates PKD-controlled vesicle biogenesis at the TGN we employed the retention using selective hooks (RUSH) assay (*Boncompain et al., 2012*). In this system, secretory cargo is retained in the ER through streptavidin-based retention and released to the secretory pathway upon biotin addition. Since TNFα was shown to be transported in Rab6 vesicles from the TGN to the plasma membrane ([*Micaroni et al., 2013*] and *Figure 8A*) and is exocytosed at FAs (personal correspondence F. Perez (Institute Curie, Paris) November, 2017), we studied whether TNFα secretion is controlled by the GEF-H1-dependent Rho signaling network. To do so, we expressed Str-KDEL-TNFα-SBP-mCherry in HeLa cells and analyzed the post-Golgi transport in fixed samples (*Figure 8B and C*) and by live cell imaging (*Figure 8D*). While TNFα-SBP-mCherry clearly left the Golgi and reached the plasma membrane after biotin addition in control cells, GEF-H1, PLCε, and PKD2/3 knockdown caused a substantial block in transport of TNFα-SBP-mCherry from the TGN to the plasma membrane (*Figure 8C and D*, *Figure 8—Video 1*). Notably, we could not observe any changes in the trafficking of TNFα between the ER and the Golgi complex (data not shown). These data clearly show that Rho-controlled PKD activation at the TGN induces the fission of Rab6-vesicles containing cargo for targeted exocytosis at FAs.

Taken together, our studies uncover a microtubule-sensitive Rho signaling network that controls Golgi complex secretory function (*Figure 9*). We propose that extracellular cues signal through this network to the Golgi complex to coordinate the formation of membrane carriers that transport secreted factors to FAs.

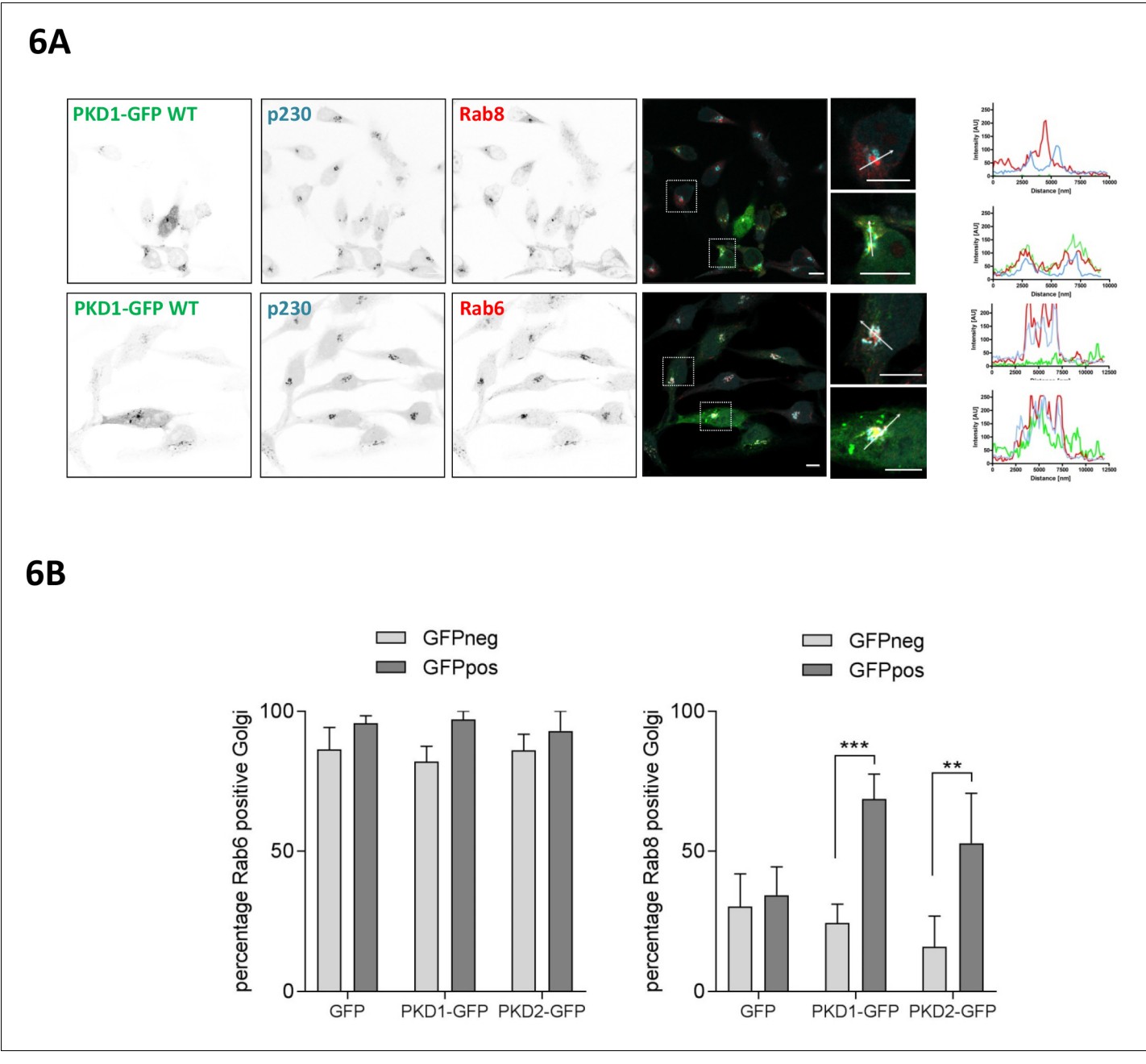

**Figure 6.** PKD recruits Rab8 to the TGN and co-localizes with Rab6. (**A**) HeLa cells were transfected with a plasmid encoding wt PKD1-GFP. One day after transfection, cells were fixed and stained for Rab8 (top panel) or Rab6 (bottom panel) plus p230 and analyzed by confocal microscopy. Scale bar 10 μm. Right panel, in the histogram (profile scan) the fluorescence intensities of the three signals along the white line are depicted. (**B**) HeLa cells were transfected with an empty GFP vector, PKD1-GFP or PKD2-GFP constructs and stained for Rab6 or Rab8 and the TGN marker protein p230. Co-localization analysis of Rab8 or Rab6 with the TGN in GFP positive and GFP negative cells was performed using Cell Profiler software as described in the material and methods section. The graph shows the mean ± SEM of three independent experiments. Significance of differences was analyzed by a two-way RM ANOVA followed by a Bonferroni's multiple comparison test. ***p=0.0004, **p=0.001.
DOI: https://doi.org/10.7554/eLife.35907.014
The following figure supplement is available for figure 6:

**Figure supplement 1.** PKD co-localizes with Rab6 and Rab8 at the Golgi compartment.
DOI: https://doi.org/10.7554/eLife.35907.015

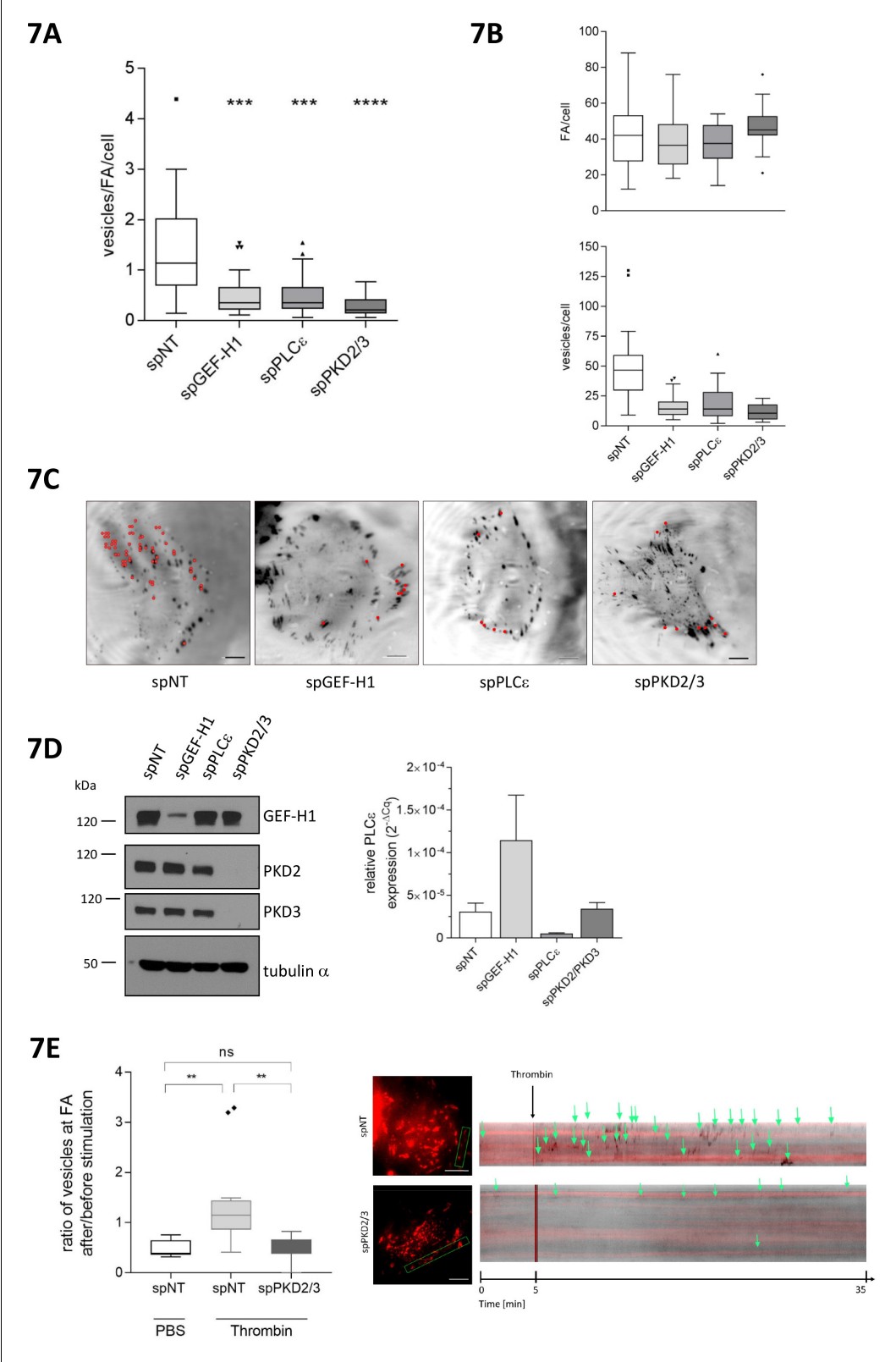

**Figure 7.** GEF-H1, PLCε and PKD are required for localized delivery of Rab6 to FAs. (**A, B, C, D, E**) Cells were transfected with spRNAs as indicated. Two days later cells were transfected with plasmids encoding GFP-tagged Rab6 and DsRed2-tagged paxillin. After 24 hr cells were analyzed at a TIRF-equipped Spinning disc microscope. The amount of Rab6-positive vesicles arriving at FAs was assessed within a two minute time interval. (**A**) The box plot shows the results of three independent experiments. Center lines show the medians; box limits indicate the 25th and 75th percentiles as

*Figure 7 continued on next page*

*Figure 7 continued*

determined by GraphPad Prism 7 software; whiskers extend 1.5 times the interquartile range from the 25th and 75th percentiles, outliers are represented by dots. n = 26, 26, 26, 20 sample points. The significance of differences was assessed by a one-way ANOVA (Kruskal Wallis test) followed by a Dunn's multiple comparison test. ****p<0.0001, ***p=0.002. (B) Top panel, number of FAs per cell. Bottom panel, number of vesicles arriving at FAs within two minutes. Center lines show the medians; box limits indicate the 25th and 75th percentiles as determined by GraphPad Prism 7 software; whiskers extend 1.5 times the interquartile range from the 25th and 75th percentiles, outliers are represented by dots. n = 26, 26, 26, 20 sample points. (C) Representative TIRF images. Red dots indicate Rab6-positive vesicles arriving at FAs within the two minute time interval. Scale bar 10 µm. (D) Left panel, silencing efficiency of GEF-H1, PKD2 and PKD3 was analyzed in lysates by immunoblotting using specific antibodies. Equal loading was verified by detection of alpha tubulin. Right panel, successful depletion of PLCε was verified by RT-qPCR. Relative expression was calculated by normalization to actin using the ΔCq method. (E) Cells were imaged 5 min prior to stimulation with thrombin or PBS (as a control). After addition of the reagent, imaging was continued for 30 min. Left panel: The box plot shows the result of three independent experiments. The vesicles arriving at FAs per minute before and after stimulation were calculated and presented as ratio '*number of vesicles at FA after/before stimulation*'. Center lines show the medians; box limits indicate the 25th and 75th percentiles as determined by GraphPad Prism 7 software; whiskers extend 1.5 times the interquartile range from the 25th and 75th percentiles, outliers are represented by dots. n = 7, 12, 10 sample points (cells analyzed). The significance of differences was assessed by a one-way ANOVA (Kruskal Wallis test) followed by a Dunn's multiple comparison test. **p<0.01, ns not significant. Right panel: Representative TIRF kymographs of spNT and spPKD2/3 cells. Arrows indicate vesicles arriving at FAs.

DOI: https://doi.org/10.7554/eLife.35907.016

The following video and figure supplement are available for figure 7:

**Figure supplement 1.** PLCε knockdown does not impair microtubule-FA interaction.
DOI: https://doi.org/10.7554/eLife.35907.017

**Figure 7—video 1.** Loss of GEF-H1, PLCε, or PKD impairs the localized delivery of Rab6-positive vesicles to FAs.
DOI: https://doi.org/10.7554/eLife.35907.018

## Discussion

Here we identify a Rho-dependent signaling network connecting microtubules with PKD-dependent vesicle fission at the TGN. Specifically, we demonstrate that GEF-H1, when released from microtubules, enhances cellular RhoA activity, which in turn promotes PKD activation at the TGN through the Rho effector PLCε, ultimately leading to the fission of Rab6-positive vesicles that move towards FAs.

The RhoGEF GEF-H1 is crucial for the coupling of microtubule dynamics to Rho GTPase activation in several biological situations such as cell polarization and motility and cell cycle regulation (reviewed in [*Birkenfeld et al., 2008*]). GEF-H1 is a well-known activator of RhoA but it can also act as a GEF for RhoB (*Arnette et al., 2016*; *Vega et al., 2012*). Release and activation of GEF-H1 can be achieved by disrupting polymerized microtubules through pharmacological compounds or, independent of microtubule disassembly, through GPCR signaling (*Meiri et al., 2014*). Here we show that both modes of GEF-H1 activation, nocodazole treatment and trypsin or thrombin-induced GPCR stimulation, initiate a Rho signaling cascade leading to PKD activation and the fission of vesicles at TGN membranes. Previous studies have demonstrated a role for GEF-H1 in vesicular trafficking: Bokoch and co-workers showed that GEF-H1 interacts with the exocyst component Sec5 leading to local RhoA activation thereby promoting exocyst assembly and exocytosis (*Pathak et al., 2012*). Additionally, the targeted trafficking of c-Src–associated endosomes to the cell periphery was reported to depend on the GEF-H1-mediated activation of endosomal RhoB (*Arnette et al., 2016*). It is thus evident that the subcellular localization of the microtubule-regulated GEF-H1 determines the spatio-temporal activation of specific Rho pools and Rho isoforms. In general, the spatial distribution of GEF-H1 is regulated by protein interactions. For example, the junctional adaptor cingulin recruits GEF-H1 to tight junctions thereby inactivating its GEF activity and inhibiting Rho signaling (*Aijaz et al., 2005*). By contrast, the interaction of GEF-H1 with Sec5 promotes RhoA activation at the exocyst complex (*Pathak et al., 2012*) and is important for GEF-H1 localization to FAs (*Wang et al., 2015*). Recently, plasma membrane associated active RhoA was demonstrated to recruit GEF-H1 thereby promoting a self-amplification loop. At the same time, active RhoA was also shown to inhibit itself by the local recruitment of the actomyosin-associated RhoGAP Myo9b (*Graessl et al., 2017*). A similar autoregulatory loop involving GEF-H1, Rho and the RhoGAP DLC3 might drive the self-limiting fission of exocytic vesicles at the TGN. Although previous reports demonstrated GEF-H1 localization to Golgi membranes (*Birkenfeld et al., 2007*; *Callow et al., 2005*), we were not able to detect endogenous or ectopically expressed GEF-H1 at the TGN. The question

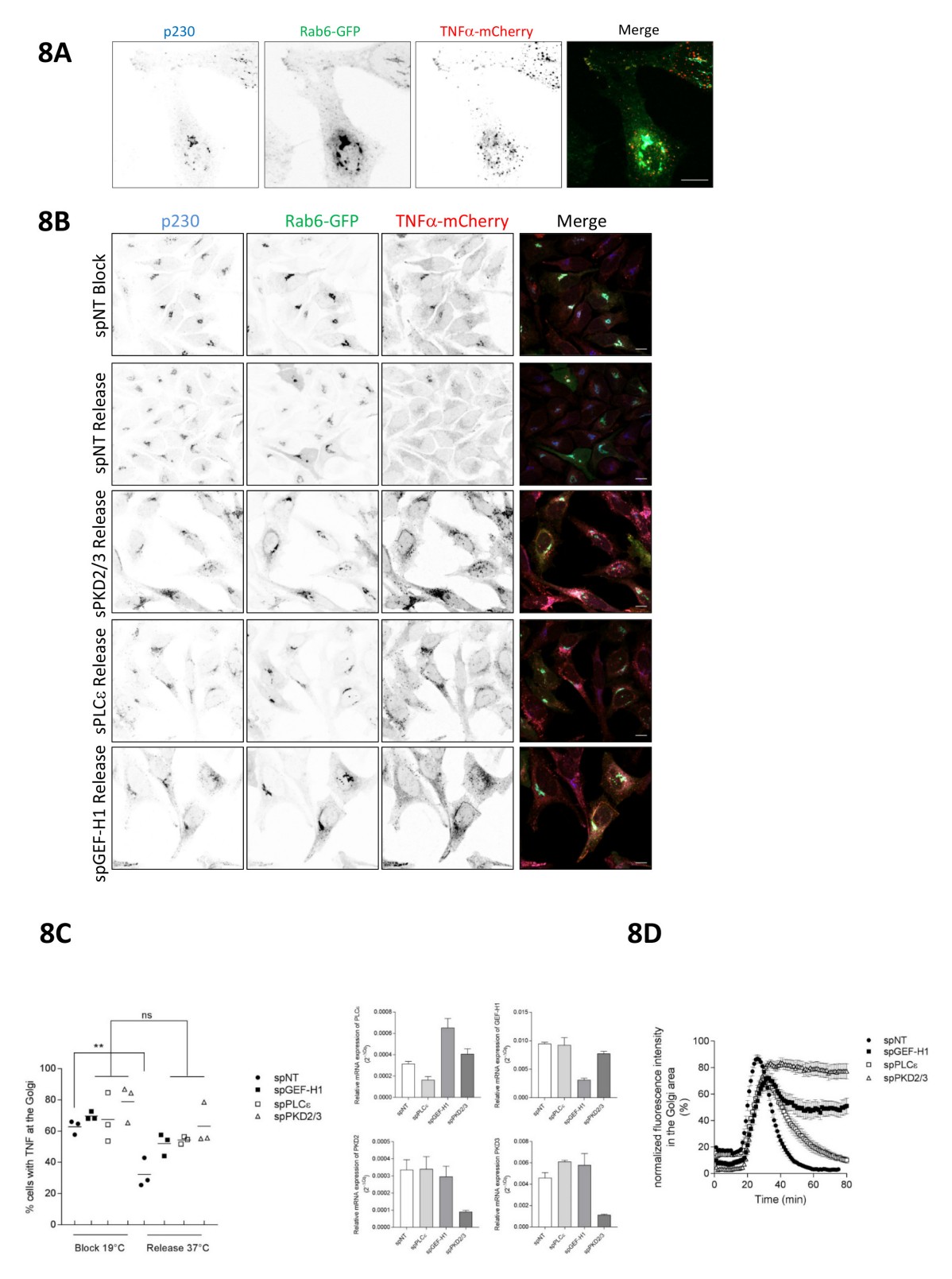

**Figure 8.** The Rho signaling pathway controls vesicle fission and cargo transport at the level of the TGN. (**A**) HeLa cells were transfected with a construct encoding Str-KDEL-TNFα-SBP-mCherry together with Rab6-GFP. 24 hr later, biotin was added for 35 min followed by fixation. Cells were stained for p230 and analyzed by confocal microscopy. Shown is a maximum intensity projection, scale bar 10 μm. (**B, C, D**) HeLa cells were transfected with spRNAs as indicated, spNT was used as a control. (**B, C**) Two days later cells were transfected with plasmids encoding Str-KDEL-TNFα-SBP-

*Figure 8 continued on next page*

*Figure 8 continued*

mCherry and Rab6-GFP and, after 24 hr, biotin was added and cells were subjected to the RUSH assay as described in the material and methods section. After fixation, cells were stained for p230. Shown are representative confocal images, scale bar 10 μm. (C) Left panel, the scatter dot blot shows the result of three independent experiments, line indicates the mean. Each dot represents one experiment with at least 190 cells analysed. The significance of differences was analyzed by a one-way ANOVA followed by a Holm-Sidak's multiple comparison test, **p=0.0065. All other comparisons were not significant. Right panel, successful depletion of the proteins was verified by RT-qPCR. Relative expression was calculated by normalization to actin using the ΔCq method. The graphs represent the mean ± SEM of three independent experiments. (D) HeLa cells were transfected with a plasmid encoding Str-KDEL-TNFα-SBP-mCherry and release from the ER was induced by biotin addition 24 hr later. The graph shows integrated fluorescence intensity in the Golgi region at each time point, corrected for background and normalized to the maximum value. Curves depict the measurement of at least 22 cells of a representative experiment. Error bars, SEM.

DOI: https://doi.org/10.7554/eLife.35907.019

The following video is available for figure 8:

**Figure 8–video 1.** Loss of GEF-H1, PLCε, or PKD blocks transport of TNFα at the Golgi complex.

DOI: https://doi.org/10.7554/eLife.35907.020

of whether GEF-H1 activates Rho directly at the TGN or whether Rho is activated elsewhere and then recruited to Golgi membranes thus remains open.

Hyperactivation of Rho has been linked to Golgi complex fragmentation (*Cole et al., 1996*; *Zilberman et al., 2011*). Zilberman and colleagues identified the Rho effector mDia1 to be responsible for the RhoA-induced Golgi fragmentation (*Zilberman et al., 2011*). Although the inhibition or loss of PKD delayed nocodazole-induced Golgi fragmentation (*Fuchs et al., 2009*), we show here that mDia1 is not responsible for PKD activation downstream of Rho. In hippocampal neurons, PKD1 activation at the Golgi occurs via a RhoA/ROCK signaling pathway and is required for the formation of Golgi outposts (*Quassollo et al., 2015*). In HeLa cells ROCK1 and 2 were dispensable for RhoA-mediated PKD activation, instead, we identified PLCε as the key Rho effector. The reason(s) underlying the differential regulation of PKD downstream of RhoA in HeLa cells and neurons is not clear at present. Within the PI-PLC family, PLCε has an outstanding role because it is an effector of the small GTPases Rho and Ras and further acts as a GEF protein for Rap1 (*Dusaban and Brown, 2015*; *Wing et al., 2003a*). Stimulated by the direct binding to RhoA, RhoB, or RhoC, PLCε hydrolyzes PtdIns(4,5)P$_2$ to generate DAG and IP3 (*Wing et al., 2003b*). In cardiac myocytes PLCε is localized to a perinuclear compartment through binding to a muscle-specific A-kinase anchoring protein β (mAKAPβ) (*Zhang et al., 2011*, *2013*) and hydrolyzes PtdIns(4)P to generate DAG and IP2 (*Zhang et al., 2013*). Subsequently, the PLCε/DAG-activated PKD translocates to the nucleus and induces the transcription of hypertrophic genes (*Zhang et al., 2013*). Interestingly, PLCε mediates sustained signaling by driving a positive feedback loop through the small GTPase Rap1. Smrcka and colleagues suggested a model in which Rap1 recruits PLCε to the Golgi complex. PLCε in turn activates Rap1, which strengthens the PLCε association with Golgi membranes thereby maintaining local DAG levels (*Smrcka et al., 2012*). Intriguingly, Gβγ signaling at the Golgi complex triggered PLCε activation in cardiac myocytes (*Zhang et al., 2013*), however, so far this has not been connected to Rho and/or PKD-dependent vesicle fission at this compartment. Based on our findings we conclude that the DAG pool produced by PLCε is also required for basal and GPCR-induced PKD activity at the TGN, thereby linking GEF-H1-controlled Rho activity with Golgi function and transport.

Thus far, DLC3 is the only RhoGAP protein that has been associated with Golgi membranes. Depletion of DLC3 resulted in Golgi fragmentation, most likely due to enhanced local Rho activity (*Braun et al., 2015*). We now provide evidence that DLC3 counteracts GEF-H1-mediated Rho and consequently PKD activation at TGN membranes. Furthermore, DLC3 is required for the integrity of the Rab8 compartment (*Braun et al., 2015*). Loss of DLC3 and therefore increased Rho activity impaired the tubule-like distribution of Rab8 and converted it to a vesicular phenotype with enhanced perinuclear localization. This is consistent with observations by Hattula et al. showing that RhoA G14V expression or nocodazole treatment redistributed Rab8 to the perinuclear region. Remarkably, we observed a similar Rab8 phenotype upon expression of active PKD1 or PKD2 demonstrating that PKD is upstream of Rab8 localization. Rab6 and Rab8 localize to the same Golgi-derived exocytic carriers, with Rab6 controlling Rab8 localization (*Grigoriev et al., 2007*; *Grigoriev et al., 2011*; *Jones et al., 1993*; *Wakana et al., 2012*). We thus hypothesize that PKD is not only required for the fission of these carriers but also responsible for the proper assembly of

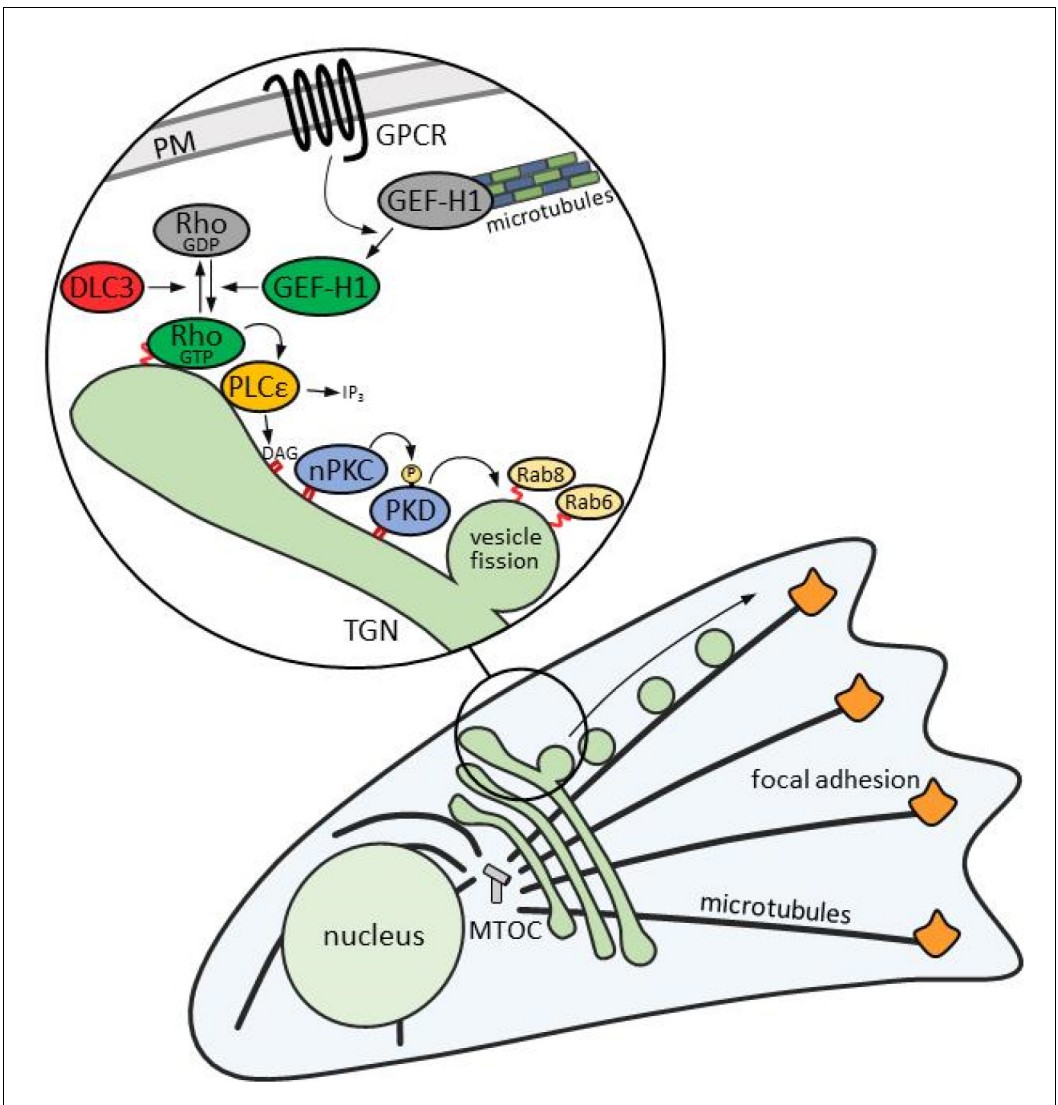

**Figure 9.** Schematic illustration of the Rho signaling pathway at the TGN. GPCR signaling activates GEF-H1 by releasing it from microtubules. GEF-H1 in turn promotes an increase in cellular RhoA activity. At the TGN, active RhoA binds to its effector PLCε, which hydrolyzes PtdIns(4)P to generate DAG and IP2. DAG recruits and activates nPKCs and PKD, which gets further activated by nPKC-mediated phosphorylation. The activation of PKD by RhoA is counterbalanced by the RhoGAP DLC3. Active PKD recruits Rab8 to TGN membranes and induces the fission of Rab6-positive vesicles containing cargo such as TNFα destined for FA-targeted transport along microtubules.
DOI: https://doi.org/10.7554/eLife.35907.021

Rab8 at the TGN. Whether PKD is upstream of Rab6 activity in this process needs to be addressed in the future.

We further provide evidence for the Rho- and PKD-dependent trafficking of TNFα in Rab6 positive vesicles. Our data are thus in line with a previous study showing that Rab6 acts on TNFα trafficking at the level of TGN exit and is required for the efficient post-Golgi transport of TNFα in inflammatory macrophages (*Micaroni et al., 2013*). In inflammatory macrophages and monocytes, TNFα secretion is induced by lipopolysaccharides (*Micaroni et al., 2013*) and further augmented by thrombin (*Hoffman and Cooper, 1995*), respectively, supporting our data that connect GPCR signaling with PKD activation at the TGN. Notably, the exocytosis of TNFα is restricted to FAs (personal correspondence F. Perez (Institute Curie, Paris), November, 2017). Likewise, Rab6-positive vesicles have been associated with the exocytosis of MT1-MMP near FAs (*Stehbens et al., 2014*) and a

recent report showed for the first time the targeted delivery of integrin-rich vesicles to FAs (*Huet-Calderwood et al., 2017*) establishing these structures as cellular hot spots of secretion.

Constitutively active RhoA was reported to augment the production of Rab6-positive vesicles (*Zilberman et al., 2011*) and Rab6-positive vesicles were shown to move along microtubules targeting FAs, thereby contributing to FA turnover during directed cell migration (*Stehbens et al., 2014*). We show here that GEF-H1, PLCε, or PKD are required for the formation and transport of these vesicles. Our results thus provide an explanation for the previously reported effects of GEF-H1 loss on cell spreading and FA turnover (*Nalbant et al., 2009*; *Vega et al., 2012*). Interestingly, extracellular matrix stiffness was found to destabilize microtubules, releasing GEF-H1, which was required for the invasion of breast cancer cells through 3D matrices (*Heck et al., 2012*). Thus, future studies should address whether microtubules, through Rho-mediated activation of Golgi-localized PKD, sense extracellular matrix density to coordinate Golgi secretory function with the increased demand for factors such as proteases or cytokines required for ECM remodeling during invasive 3D migration.

# Materials and methods

**Key resources table**

| Reagent type (species) or resource | Designation | Source or reference | Identifiers | Additional information |
|---|---|---|---|---|
| Cell line (human) | HEK293T | ATCC | ATCC Cat# CRL-3216, RRID:CVCL_0063 | |
| Cell line (human) | Flp-In T-Rex-293 | Thermo Fisher Scientific | RRID:CVCL_U427 | |
| Cell line (human) | Flp-In T-Rex-HeLa | other | | generated by Elena Dobrikova and Matthias Gromeier, Duke University Medical Center, Durham, NC, USA |
| Cell line (human) | HeLa | ATCC | ATCC Cat# CRM-CCL-2, RRID:CVCL_0030 | |
| Recombinant DNA reagent | pEGFP-N1-PKD1 wt/K612W | DOI: 10.1038/ncb1289 | | |
| Recombinant DNA reagent | pEGFP-N1-PKD2 | DOI: 10.1038/ncb1289 | | |
| Recombinant DNA reagent | pCMV5-EGFP-GEF -H1 wt | DOI: 10.1038/ncb773 | | |
| Recombinant DNA reagent | pCMV5-EGFP-GEF -H1 C53R | DOI: 10.1038/ncb773 | | |
| Recombinant DNA reagent | pTriEx-RhoA FLARE.sc | DOI: 10.1038/nature04665 | Addgene #12150 | |
| Recombinant DNA reagent | mRuby-Golgi-7 | | Addgene #55865 | |
| Recombinant DNA reagent | Rab6a-GFP | other | | provided by Francis Barr, Oxford University |
| Recombinant DNA reagent | DsRed2-Paxillin | DOI: 10.1038/ncb1094 | | |
| Recombinant DNA reagent | G-PKDrep | doi: 10.1111/j.1600–0854.2009.00918.x | | |
| Recombinant DNA reagent | Clip170-GFP | other | | provided by Niels Galjart, Erasmus Medical Center, Rotterdam |
| Recombinant DNA reagent | pcDNA5/FRT/TO-EGFP -GEF-H1 wt/C53R | this paper | | |
| Recombinant DNA reagent | pECFP-Endo/RhoB | Clontech | Clontech #6934–1 | |

*Continued on next page*

*Continued*

| Reagent type (species) or resource | Designation | Source or reference | Identifiers | Additional information |
|---|---|---|---|---|
| Recombinant DNA reagent | pECFP-Endo/Rhob Q63L | this paper | | site-directed mutagenesis using pECFP-RhoB as a template |
| Recombinant DNA reagent | pEGFP-RhoB | this paper | | subcloned from pECFP-Endo/RhoB |
| Recombinant DNA reagent | pEGFP-RhoB Q63L | this paper | | subcloned from pECFP-Endo/RhoB Q63L |
| Recombinant DNA reagent | pEGFP-RhoA | DOI: 10.1242/jcs.163857 | | |
| Recombinant DNA reagent | pEGFP-RhoA Q63L | DOI: 10.1242/jcs.163857 | | |
| Recombinant DNA reagent | pcDNA3.1-HA-RhoA Q63L | DOI: 10.1242/jcs.163857 | | |
| Recombinant DNA reagent | pcDNA3.1-HA-RhoA | DOI: 10.1242/jcs.163857 | | |
| Recombinant DNA reagent | Str-KDEL-TNFa-SBP-mCherry | | Addgene #65279 | |
| Recombinant DNA reagent | pOG44 | Thermo Fisher Scientific | Thermo Fisher Scientific V600520 | |
| Antibody | p230 | BD Biosciences | BD Biosciences Cat# 611280, RRID:AB_398808 | |
| Antibody | GFP | Roche | Sigma-Aldrich Cat# 11814460001, RRID:AB_390913 | |
| Antibody | TGN46 | Bio-Rad/AbD Serotec | Bio-Rad/AbD Serotec Cat# AHP500, RRID:AB_324049 | |
| Antibody | alpha-tubulin | Millipore | Millipore Cat# 05–829, RRID:AB_310035 | |
| Antibody | DLC3 | Santa Cruz Biotechnology | Santa Cruz Biotechnology Cat# sc-166725, RRID:AB_2197829 | |
| Antibody | PKD1 | Cell Signaling | Cell Signaling Technology Cat# 2052, RRID:AB_2268946 | |
| Antibody | PKD2 | Cell Signaling | Cell Signaling Technology Cat# 8188S, RRID:AB_10829368 | |
| Antibody | PKD3 | Cell Signaling | Cell Signaling Technology Cat# 5655S, RRID:AB_10695917 | |
| Antibody | GEF-H1 | Cell Signaling | Cell Signaling Technology Cat# 4076, RRID:AB_2060032 | |
| Antibody | phospho-PKD (Ser744/748) | Cell Signaling | Cell Signaling Technology Cat# 2054S, RRID:AB_2172539 | |
| Antibody | Rab8 | Cell Signaling | Cell Signaling Technology Cat# 6975S, RRID:AB_10827742 | |
| Antibody | Rab6 | Cell Signaling | Cell Signaling Technology Cat# 9625S, RRID:AB_10971791 | |
| Antibody | pS910 | DOI: 10.1083/jcb.200110047 | | |

*Continued on next page*

*Continued*

| Reagent type (species) or resource | Designation | Source or reference | Identifiers | Additional information |
|---|---|---|---|---|
| Antibody | ERK1/2 | Cell Signaling | Cell Signaling Technology Cat# 9107S, RRID:AB_10695739 | |
| Antibody | MEK1/2 | Cell Signaling | Cell Signaling Technology Cat# 8727, RRID:AB_10829473 | |
| Antibody | pERK1/2 (Thr202/Tyr204) | Cell Signaling | Cell Signaling Technology Cat# 4094S, RRID:AB_10694057 | |
| Antibody | pMEK1/2 (Ser2017/221) | Cell Signaling | Cell Signaling Technology Cat# 9154, RRID:AB_2138017 | |
| Antibody | ROCK1 | Millipore | Millipore Cat# 04–1121, RRID:AB_1977472 | |
| Antibody | ROCK2 | BD Biosciences | BD Biosciences Cat# 610623, RRID:AB_397955 | |
| Antibody | pS294 | DOI: 10.1038/ncb1289 | | IF 1:750 |
| Antibody | HRP goat anti-rabbit | Dianova | Jackson ImmunoResearch Labs Cat# 111-035-144, RRID:AB_2307391 | |
| Antibody | HRP goat anti-mouse | Dianova | Jackson ImmunoResearch Labs Cat# 115-035-062, RRID:AB_2338504 | |
| Antibody | Alexa Fluor labelled secondary antibodies | Thermo Fisher Scientific | Thermo Fisher Scientific | |
| Sequence-based reagent | ON-Targetplus smartpools | Dharmacon | Dharmacon | |
| Sequence-based reagent | Silencer select siPLCe | Thermo Fisher Scientific | Thermo Fisher Scientific s27660 | |
| Sequence-based reagent | Silencer select siGEF-H1 | Thermo Fisher Scientific | Thermo Fisher Scientific s17546 | |
| Sequence-based reagent | Quantitect primers for RT-PCR | Qiagen | Qiagen | |
| Peptide, recombinant protein | Trypsin | Thermo Fisher Scientific | Thermo Fisher Scientific 15090046 | |
| Peptide, recombinant protein | Thrombin | Millipore | Millipore 605195 | |
| Commercial assay or kit | QuantiTect SYBR Green RT-PCR Kit | Qiagen | Qiagen 204243 | |
| Commercial assay or kit | Rneasy plus Kit | Qiagen | Qiagen 74104 | |
| Chemical compound, drug | Nocodazole | Sigma-Aldrich | Sigma-Aldrich M1404 | |
| Chemical compound, drug | CRT0066101 | Tocris Bioscience | Tocris Bioscience 4975 | |
| Chemical compound, drug | H1152 | Enzo Life Science | Enzo Life Science ALX-270–423 M001 | |
| Chemical compound, drug | UO126 | Cell Signaling | Cell Signaling #9903 | |
| Chemical compound, drug | Blasticidin | Thermo Fisher Scientific | Thermo Fisher Scientific R21001 | |
| Chemical compound, drug | Hygromycin B | Thermo Fisher Scientific | Thermo Fisher Scientific 10687010 | |

*Continued on next page*

*Continued*

| Reagent type (species) or resource | Designation | Source or reference | Identifiers | Additional information |
|---|---|---|---|---|
| Chemical compound, drug | Doxycyclin | Sigma-Aldrich | Sigma-Aldrich D9891 | |
| Chemical compound, drug | Zeocin | Thermo Fisher Scientific | Thermo Fisher Scientific R25001 | |
| Chemical compound, drug | Collagen R | Serva | Serva 47254 | |
| Chemical compound, drug | Biotin | Sigma-Aldrich | Sigma-Aldrich B4501 | |
| Chemical compound, drug | Blocking reagent | Roche | Roche 11096176001 | |
| Software, algorithm | Cell Profiler | http://cellprofiler.org | CellProfiler Image Analysis Software, RRID:SCR_007358 | |
| Software, algorithm | GraphPad Prism | GraphPad Prism (https://graphpad.com) | Graphpad Prism, RRID:SCR_002798 | |
| Software, algorithm | Image Studio Lite 4.0 | https://www.licor.com/bio/products/software/image_studio_lite/?utm_source=BIO+Blog&utm_medium=28Aug13post&utm_content=ISLite1&utm_campaign=ISLite | Image Studio Lite, RRID:SCR_014211 | |

## Reagents, plasmids, antibodies

The plasmid encoding G-PKDrep was described previously (*Fuchs et al., 2009*). Cherry-tagged G-PKDrep was generated by replacing GFP with mCherry. Plasmids encoding GFP-tagged PKD1 wt, kd PKD1 K612W, and GFP-tagged PKD2 were described previously (*Hausser et al., 2005*). The RhoA WT biosensor pTriEx-RhoA FLARE.sc was purchased from Addgene (Addgene plasmid # 12150; [*Pertz et al., 2006*]). mRuby-Golgi-7 (N-terminal 81 amino acids of human $\beta-1,4$-galactosyl-transferase) was a gift from Michael Davidson (Addgene plasmid # 55865). The plasmid encoding Rab6a-GFP was provided by Francis Barr (Oxford University), DsRed2-paxillin was received from Rick Horwitz (University of Virginia). Plasmids encoding GFP-tagged GEF-H1 wt and C53R were provided by Perihan Nalbant (University of Duisburg-Essen) and are described in detail in (*Krendel et al., 2002*). pEGFP-RhoA Q63L and pcDNA3.1-HA-RhoA constructs were described elsewhere (*Braun et al., 2015*). Clip170-GFP was a kind gift from N. Galjart (Erasmus Medical Center, Rotterdam, The Netherlands). The pECFP-RhoB/Endo vector was purchased from Clontech (Mountainview, CA, USA, Clontech plasmid #6934–1). pEGFP-RhoB was generated by subcloning of RhoB into the pEGFP vector. The mutation Q63L was inserted by site-directed mutagenesis using the primers RhoB Q63L FP 5' gtgggacacagctggcctggaggactacgaccgc 3' and RhoB Q63L RP 5' gcggtcgtagtcctccaggccagctgtgtcccac 3' (purchased from Eurofins Genomics, Ebersberg, Germany). To generate the inducible GEF-H1 expression plasmids pcDNA5/FRT/TO-EGFP-GEF-H1 wt and pcDNA5/FRT/TO-EGFP-GEF-H1 C53R, the cDNA encoding GFP-GEF-H1 was excised from the pCMV5-EGFP-GEF-H1 plasmid with *EcoRI and HindIII,* and, after DNA blunting, ligated with pcDNA5/FRT/TO digested with *EcoRV*. Integrity of the construct was verified by sequencing. The pSer294-specific rabbit polyclonal antibody used for detection of G-PKDrep phosphorylation was described before (*Fuchs et al., 2009*). The antibody specific for PKD1 autophosphorylation at serine 910 has been described elsewhere (*Hausser et al., 2002*). Commercially available antibodies used were as follows: TGN46-specific sheep antibody was from Bio-RAD. The following antibodies were from Cell Signaling Technologies (Danvers, MA, USA): anti-Rab8, anti-Rab6, anti-PKD2, anti-PKD3, anti-GEF-H1 rabbit monoclonal antibodies and anti-phospho-PKD (Ser744/748) and anti-PKD1 rabbit antibodies, mouse mAb ERK1/2 (3A7), rabbit mAb MEK1/2 (D1A5), rabbit mAb pERK1/2 (Thr202/Tyr204) (D13.14.4E), rabbit mAb pMEK1/2 (Ser217/221) (41G9). The ROCK1-specific rabbit monoclonal antibody EPR638Y was from Merck Chemicals, anti-ROCK2 mouse monoclonal antibody clone 21 was from BD Biosciences, monoclonal mouse anti-DLC3 (E-2) (Santa Cruz Biotechnology, Dallas, Texas, USA), anti–tubulin α mouse monoclonal antibody (Merck Chemicals GmbH, Darmstadt, Germany), anti-

p230 (BD Biosciences, Heidelberg, Germany), and anti-GFP mouse monoclonal antibody (Roche Diagnostics). Secondary antibodies used were Alexa405, Alexa488, Alexa546, or Alexa633 coupled goat anti–mouse and anti–rabbit immunoglobulin G (IgG) (Life Technologies, Carlsbad, CA, USA), and horseradish peroxidase (HRP) coupled goat anti–mouse and anti–rabbit IgG (Dianova, Hamburg, Germany). Alexa633-coupled phalloidin was obtained from Life Technologies. Nocodazole was obtained from Sigma-Aldrich, trypsin was from Thermo Fisher Scientific, thrombin from Merck Millipore, UO126 was obtained from Cell Signaling Technologies. H1152 was from Enzo Life Science (Farmingdale, NY, USA). CRT0066101 was from Tocris Bioscience (Bristol, UK).

## Protein extraction of cells and Western blotting

Whole cell extracts were obtained by solubilizing cells in lysis buffer (20 mM Tris pH 7.4, 150 mM NaCl, 1% Triton X-100, 1 mM EDTA, 1 mM ethylene glycol tetra acetic acid (EGTA), plus Complete protease inhibitors and PhosSTOP (Roche Diagnostics, Basel, Switzerland)). Whole cell lysates were clarified by centrifugation for 15 min at 16,000 g and 4°C. Equal amounts of protein were loaded on 10% polyacrylamide gels or were run on NuPage Novex 4–12% Bis-Tris or 3–8% Tris-Acetate gels (Life Technologies) and blotted onto nitrocellulose membranes using the iBlot device (Life Technologies). Membranes were blocked for 30 min with 0.5% (v/v) blocking reagent (Roche Diagnostics) in PBS containing 0.05% (v/v) Tween-20. Membranes were incubated with primary antibodies overnight at 4°C, followed by 1 hr incubation with HRP-conjugated secondary antibodies at room temperature. Proteins were visualized using an enhanced chemiluminescence detection system (Thermo Fisher Scientific, Waltham, MA, USA). For quantitative Western Blotting chemiluminescence was detected at a depth of 16-bit in the linear detection range of an Amersham Imager 600 equipped with a 3.2 megapixel super-honeycomb CCD camera fitted with a large aperture f/0.85 FUJINON lens. Special care was taken not to overexpose in order to guarantee accurate quantifications. For all proteins, at least three independent membranes were analyzed. Densitometry was performed using Image Studio Lite 4.0 (Li-COR Biosciences, Bad Homburg, Germany). For each protein, the integrated density of the signal was measured, corrected for background signals and adjusted to loading controls.

## Cell culture and transfection

HeLa and HEK293T cells were maintained in RPMI 1640 medium supplemented with 10% FCS. Cell lines were authenticated using Multiplex Cell Authentication by Multiplexion (Heidelberg, Germany) as described recently (Castro et al., 2013). The SNP profiles matched known profiles or were unique. Cells were tested negative for mycoplasma contamination using MycoAlert (Lonza, Switzerland). For transient plasmid transfections, HEK293T cells were transfected with TransIT-293 (Mirus Bio, Madison, WI, USA). HeLa cells were transfected with TransIT-HeLaMONSTER (Mirus Bio) or in case of RUSH experiments with FuGENE HD transfection reagent (Promega, Madison, WI, USA) according to the manufacturer's instructions. In the case of siRNA oligonucleotides, HEK293T and HeLa cells were transfected with Lipofectamine RNAimax (Life Technologies) according to the manufacturer's instructions. After 48 hr, siRNA-transfected cells were further transfected with plasmid DNA and analyzed 24 hr later. As a negative control (termed spNT), ON-TARGETplus non-targeting control pool D-001810–10 from Dharmacon (Lafayette, CO, USA) was used. siRNAs used were: spDLC3 (siGENOME SMARTpool human STARD8 M-010254), spmDia1 (ON-Target plus SMARTpool human DIAPH1, L-010347), spGEF-H1 (ON-Target plus SMARTpool human ARHGEF2 L-009883), spPLCε(ON-Target plus SMARTpool human PLCE1 J-004201), spPKD2 (ON-Target plus SMARTpool human PRKD2 L-004197, spPKD3 (ON-Target plus SMARTpool human PRKD3 L-005029), spROCK1 (ON-Target plus SMARTpool human ROCK1 L-003536), and spROCK2 (ON-Target plus SMARTpool human ROCK2 L-004610). All smartpools were obtained from Dharmacon. The following single siRNAs were obtained from Thermo Fisher Scientific: siPLCε (Silencer Select s27660) and siGEF-H1 (Silencer Select s17546). Flp-In T-REx-293 cells (Life Technologies) and Flp-In T-REx-HeLa cells (generated by Elena Dobrikova and Matthias Gromeier, Duke University Medical Center, Durham, NC, USA) were grown in DMEM containing 10% FCS, 100 µg/ml zeocin and 15 µg/ml (293) or 10 µg/ml blasticidin (HeLa). These cells stably express the Tet repressor, contain a single Flp Recombination Target (FRT) site and were used to generate the Flp-In-T-REx-EGFP-GEF-H1- lines. Cells were cotransfected with pcDNA5/FRT/TO-EGFP-GEF-H1 wt or C53R and the Flp recombinase expression plasmid pOG44 at a ratio of 1:10 and then selected with 100 µg/ml (293) or 500 µg/ml (HeLa)

hygromycin B. Induction of protein expression with doxycycline was at 10 ng/ml. Treatment of cells with nocodazole was at 5 µg/ml, trypsin and thrombin were used at 10 nM.

## Quantitative one-step real-time PCR

RNA was isolated from cells using the RNeasy plus Kit (Qiagen, Hilden, Germany) following the manufacturers' instructions. 100 ng RNA were used for the real-time PCR reaction using the QuantiTect SYBR Green RT-PCR Kit from Qiagen following the manufacturer's instructions. Analysis was performed using the CFX96 Touch Real-Time PCR Detection System (Bio-RAD, Hercules, CA, USA). The following QuantiTect primers were used: Hs_GAPDH_2_SG , Hs_PLCE1_1_SG, Hs_ARHGEF2_1_SG, Hs_PRKD2_1_SG, Hs_PRKD3_1_SG, and Hs_ACTB_2_SG (all obtained from Qiagen).

## Immunofluorescence staining and confocal microscopy

Cells grown on glass coverslips coated with 2,5 µg/ml collagen R (Serva, Heidelberg, Germany) were fixed for 15 min with 4% (v/v) paraformaldehyde. After washes in PBS, cells were incubated for 5 min with 1 M glycine in PBS and permeabilized for 2 min with 0.1% (v/v) Triton X-100 in PBS. Blocking was performed with 5% (v/v) bovine serum (PAN) in PBS for 30 min. Fixed cells were incubated with primary antibodies diluted in blocking buffer for 2 hr at room temperature. Following three washing steps with PBS, cells were incubated with Alexa-Fluor-(488, 546 or 633)-labeled secondary antibodies in blocking buffer for 1 hr at room temperature. Nuclei were counterstained with DAPI and mounted in ProLong Gold Antifade Reagent (Thermo Fisher Scientific). All samples were analyzed at room temperature using a confocal laser scanning microscope (LSM 710, Carl Zeiss) equipped with a Plan Apochromat 63x/1.40 DIC M27 (Carl Zeiss, Jena, Germany) oil-immersion objective. GFP was excited with the 488 nm line of an Argon laser, its emission was detected from 496 to 553 nm. Alexa546 was excited with a 561 nm DPSS laser, its emission was detected from 566 to 622 nm. In case of Phalloidin-Alexa633 staining, Alexa633 was excited with a 633 nm HeNe laser, its emission was detected from 638 to 740 nm. Image acquisition for G-PKDrep ratiometric imaging was done as follows: z-stacks of 0.5 µm intervals were acquired throughout the cell and maximum intensity projections were calculated. GFP and Alexa546 channels were hereby acquired with the same pinhole setting that was adjusted to 1 AU in the Alexa 546 channel. Laser powers were adjusted to prevent fluorophore saturation and identical photomultiplier tube and laser settings were maintained throughout the whole experiment. Image processing and analysis was performed with Zen black 2.1 software. Regions of interest of identical Golgi areas of reporter expressing cells were selected in the GFP channel, mean pixel intensity values of the selected areas in both channels were measured and the Alexa546 to GFP ratio was calculated. In the case of the Cherry-tagged G-PKDrep construct, staining of the pSer294 signal was performed using an Alexa633-conjugated secondary antibody. Image acquisition was done in the Cherry and Alexa633 channels as described above. Alexa633 pixel intensity was corrected for background signals.

## FRET imaging

FRET studies with the RhoA activity biosensor were performed on a Zeiss Axio Observer Spinning Disc microscope equipped with a Plan-Apochromat 63x/1.4 Oil DIC objective and a Photometrix Evolve 512 EMCCD camera. CFP and FRET channels were acquired with a 445 nm diode excitation laser combined with 485/30 nm (CFP) and 562/45 nm (FRET) emission filters. mRuby was acquired using a 561 nm diode excitation laser combined with a 600/50 nm emission filter. Images were acquired at 37°C every 60 s for one hour at a resolution of 512 × 512 Px. Focus was stabilized using the Definite Focus 2 autofocus device. Nocodazole was added 10 min after start of image acquisition. Image analysis was done with Zen blue 2.3 software. After background subtraction the mean FRET/CFP (Y/C) emission ratio was calculated for every time point. In order to measure RhoA activity specifically at the Golgi complex, a ROI was drawn around the Golgi compartment (identified through the mRuby-Golgi7 signal). The mean FRET/CFP emission ratio of the ROI of multiple cells was normalized to its values in unstimulated state and plotted against the time. In parallel, ratio images that reflect the RhoA activation state throughout the cell were generated.

## TIRF microscopy

TIRF images were acquired on the same Zeiss Axio Observer microscope stand used for FRET imaging, additionally equipped with a motorized TIRF illuminator (Laser TIRF 3), an EMCCD camera (Photometrics Evolve 512) and an alpha Plan-Apochromat 100x/1.46 NA Oil objective at 37°C and 5% $CO_2$ in phenolred-free RPMI supplemented with 10% FCS. In case of the basal localized delivery experiment, TIRF-images of GFP-Rab6 and DsRed2-Paxillin were acquired every second for a time interval of two minutes. Image processing was done with the Zen 2.1 blue software. All channels were filtered with a Low-pass filter followed by an unsharp-masking in order to reduce pixel noise and enhance the contrast of dim structures. In case of the thrombin stimulation, cells were serum starved for 30 min prior to image acquisition. TIRF images were acquired every two seconds for a time interval of 35 min. Thrombin was added after 5 min. Control cells received PBS. Kymographs were generated using Zen 2.3 blue software. For each cell, three to five kymographs were generated. At least 10 focal adhesions per cell were analyzed for vesicle arrival.

## 3D time lapse

3D time lapse studies were performed on the Zeiss Axio Observer Spinning Disc system equipped with a Plan-Apochromat 63x/1.4 Oil DIC objective and two Axiocam 503 mono CCD cameras. GFP-Rab6A and DsRed2-Paxillin images were acquired simultaneously in Dual-camera mode using a 560 nm beamsplitter. The following excitation lasers and emission filters were used: GFP, 488 nm diode laser, 525/50 nm filter, RFP, 561 nm (RFP) diode laser, 600/50 nm filter. Z-Stacks with a 500 nm interval were acquired every second for two minutes. 3D time lapse sequences were rendered using the Zen blue 2.1 3d VisArt module.

## Golgi localization analysis

Images for Golgi localization analysis were acquired on the LSM710 microscope equipped with an EC-Plan Neofluar 40x/1.30 NA Oil DIC objective. GFP and Alexa 546 images were acquired with the same settings as described confocal microscopy section. Quantitative image analysis was done with CellProfiler version 2.1.1 (*Carpenter et al., 2006*; *Kamentsky et al., 2011*). Ten confocal overview images (zoom factor 0.6) were analyzed per condition. The Golgi complex and the cytoplasm were segmented using the p230 staining and its background, respectively. The mean intensity of the GFP signal was measured under both segmentation masks and the Golgi/Cytoplasm ratio was generated for every cell representing the PKD1kd-GFP distribution throughout the cell. GFP-negative cells were excluded from the analysis.

## Kymography

Microtubuli polymerization studies were performed on the Zeiss Axio Observer Spinning Disc system equipped with a Plan-Apochromat 63x/1.4 Oil DIC objective and two Axiocam 503 mono CCD cameras. Clip170-GFP and DsRed2-Paxillin were acquired sequentially every second for 2 min using an automated emission filter wheel for CSU-X1. The following excitation lasers and emission filters were used: GFP, 488 nm diode laser, 525/50 nm filter, RFP, 561 nm (RFP) diode laser, 600/50 nm filter. Kymographs were generated in the Zen blue 2.1 software.

## Golgi co-localization analysis

For the Rab6 and Rab8 Golgi co-localization analysis, HeLa control- and siRNA knockdown cells, overexpressing GFP or a GFP-tagged version of PKD1 or PKD2 were plated on coverslips, fixed and stained for p230 and Rab6 or Rab8 according to the protocol described in the immunofluorescence staining section. For image acquisition, overview images with maximal field of view were taken at a resolution of 2048 × 2048 on the LSM710 microscope equipped with a Plan-Apochromat 63x/1.40 Oil DIC objective. GFP, Alexa 546 and Alexa 633 channels were acquired with the same settings as described in the confocal microscopy section. The quantitative object-based co-localization analysis was done using CellProfiler version 2.1.1 (*Carpenter et al., 2006*; *Kamentsky et al., 2011*). In brief, Golgi complexes were segmented and measured for GFP and Rab6 or Rab8 staining intensities. Each Golgi was then classified in order to be positive or negative for PKD and/or Rab6 and Rab8 using a threshold based cut-off line. For co-localization analysis of RhoA and RhoB with Rab6, HeLa cells were transiently transfected with a GFP-tagged version of RhoA/B wt or Q63L. Cells were

plated on coverslips, fixed and stained for Rab6 (Alexa 633) as described in the immunofluorescence staining section. Optical slices were acquired at the LSM710 confocal microscope in all four channels as described in the confocal microscopy section. The quantitative co-localization analysis was performed with CellProfiler software (version 3.0.0). First, single cells were segmented based on the GFP image. Afterwards, Manders' coefficient (Costes automated threshold) was determined for the GFP and Alexa633 overlap in each cell.

## RUSH (Retention using selective hooks) trafficking assay

HeLa cells were maintained in biotin-free DMEM supplemented with 10% FCS. SiRNA-transfected cells were plated on glass coverslips and transiently co-transfected with a plasmid encoding Str-KDEL-TNFα-SBP-mCherry (*Boncompain et al., 2012*) together with Rab6-GFP at a ratio of 4:1 using FuGENE HD transfection reagent (Promega) according to the manufacturer´s protocol. After biotin-induced release of the TNFα-mCherry cargo from the ER, cells were incubated for 60 min at 19°C to block cargo exit from the Golgi complex. Cargo was released from the Golgi through a temperature shift to 37°C using pre-warmed medium. Cells were fixed at time points 0 and 35 min after release. Samples were stained for the TGN marker p230 using an Alexa405-labelled secondary antibody. Images were acquired at the AxioObserver SD described above. Z-Stacks were acquired with the 63x objective in 405, GFP and Cherry channels at a 500 nm interval and maximum intensity projections were calculated. Only cells expressing Rab6-GFP and TNFα-mCherry at a moderate level were analyzed. The trafficking status of TNFα-mCherry was analyzed and thereby each cell was classified in two categories: 1. 'Golgi' cells, in which the TNF-cargo remained partially at the Golgi compartment. 2. 'Traffic' cells, in which the TNF-cargo exited the Golgi completely. For live cell imaging siRNA-transfected cells were plated in 35 mm glass bottom petri dishes and transiently transfected with the Str-KDEL-TNFα-SBP-mCherry plasmid using FuGENE HD transfection reagent. Cells were imaged in FluoroBrite DMEM (Thermo Fisher Scientific) supplemented with 10% FCS. Time series of images were acquired at the AxioObserver SD described above equipped with an Evolve 512 EMCCD camera (Photometrics, Tucson, AZ, USA). Z-Stacks were acquired with the 63x objective in the Cherry channel at a 500 nm interval every minute for 2 hr. Cargo release from the ER was induced by the addition of biotin at 40 µM after 10 cycles. For image analysis maximum intensity projections were calculated and the mean intensity in the Golgi region was measured over time. The obtained intensities were normalized to the highest value that was considered as the starting point to observe the Golgi traffic to the plasma membrane for each cell.

## Acknowledgements

We are grateful to Attila Ignacz (Eötvös Lorand University Budapest, Hungary) for help in designing *Figure 9*. We would like to thank Franck Perez and Gaelle Boncompain (Institute Curie, Paris, France) for providing the RUSH constructs and giving helpful advices. The lab of Angelika Hausser is supported by grants from the German cancer aid (DKH 111941), German research foundation (DFG HA-3557/11–2) and the Volkswagen Foundation (88373).

## Additional information

### Funding

| Funder | Grant reference number | Author |
|---|---|---|
| Deutsche Krebshilfe | DKH 111941 | Angelika Hausser |
| Deutsche Forschungsge-meinschaft | DFG HA-3557/11–2 | Angelika Hausser |
| Volkswagen Foundation | 88373 | Angelika Hausser |

The funders had no role in study design, data collection and interpretation, or the decision to submit the work for publication.

## Author contributions
Stephan A Eisler, Investigation, Visualization, Writing—original draft; Filipa Curado, Gisela Link, Sarah Schulz, Melanie Noack, Maren Steinke, Investigation, Visualization; Monilola A Olayioye, Resources, Writing—review and editing; Angelika Hausser, Conceptualization, Formal analysis, Funding acquisition, Visualization, Writing—original draft, review and editing

## Author ORCIDs
Angelika Hausser (iD) http://orcid.org/0000-0002-4102-9286

## Decision letter and Author response
Decision letter https://doi.org/10.7554/eLife.35907.024
Author response https://doi.org/10.7554/eLife.35907.025

# Additional files

## Supplementary files
• Transparent reporting form
DOI: https://doi.org/10.7554/eLife.35907.022

## Data availability
All data generated or analysed during this study are included in the manuscript and supporting files.

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
