## [Decision Letter]

Thank you for submitting your article "A Rho signaling network links microtubules to PKD controlled carrier transport to focal adhesions" for consideration by *eLife*. Your article has been reviewed by three peer reviewers, and the evaluation has been overseen by Suzanne Pfeffer as the Reviewing Editor and Vivek Malhotra as the Senior Editor. The reviewers have discussed the reviews with one another and the Reviewing Editor has drafted this decision to help you prepare a revised submission.

This manuscript investigates the RhoA-dependent activation of PKD at the Golgi. While RhoA has been observed at the Golgi over a decade, it has not been clear what this small GTPase is doing at this location. The current work establishes a RhoA-PLCε-PKD signaling cascade that operates at the Golgi and is important for maintaining Golgi structure and for the export of Rab6 carriers from this organelle. The work is important but the following issues should be addressed before publication can be recommended.

1) The most important concern is whether the pathway studied is truly physiological. For example, the experiment in Figure 4C, in which PKD activity is assessed under basal conditions (i.e., not treated with nocodazole, trypsin, or GTP-locked RhoA), indicates that PLCε plays only a very minor role in basal PKD activity. Does PAR activation actually alter secretion level (e.g. of TNFα)? If so, then to what extent is the level changed? And, what is a likely physiological context for this regulation? There are many papers showing the impact of altering intracellular signaling factors (including the ones studied here) on secretory pathway organization and/or function. What is missing is a demonstration that these changes occur in a physiologically relevant signaling condition with the expected changes in output (i.e. added ligand X causing a Y-fold change in localized secretion of Z). Here, we believe it would greatly improve the impact of the study to at least quantify PKD-dependent secretion of TNFα in response to activation of a PAR (one wishes something more selective than trypsin could be used for the activation).

2) The FRET experiment in Figure 2D shows that depletion of GEF-H1 results in a reduction in the total cellular activity of RhoA (lower panel). Therefore, GEF-H1 does not seem to be a Golgi-specific regulator of RhoA. In the upper row of the same panel, treatment with nocodazole also increases total cellular RhoA FRET (i.e. activity). It is hard to see the Golgi in this image, most likely because of fragmentation induced my microtubule depolymerization. It is hard to see Golgi-localized FRET signals in the images provided, even zoomed in. Based on this experiment it is not possible to say that GEF-H1 regulates RhoA activity at the Golgi. Rather, GEF-H1 regulates the total cellular pool of active RhoA, including the Golgi pool.

3) Is GEF-H1 localized at the Golgi? Also, in Figure 1C, the GFP-RhoA-Q63L image shows that this fusion protein localizes to the Golgi in only a subset of the cells. The extent of Golgi localization should be quantified and reported. It appears that the corresponding RhoB construct has much better Golgi localization.

4) The G-PKDrep reporter is heavily utilized in this work, and has been previously validated. However, when new treatments are used that have not previously been validated, it is important to verify that the reporter is indeed reporting PKD activity, as it is easy to imagine that another kinase could phosphorylate the reporter. For example, how do we know that GTP-locked RhoA does not activate another kinase that is able to phosphorylate the G-PKDrep reporter? Or that Trypsin treatment doesn't activate another kinase that will phosphorylate the reporter? Therefore, for any new experimental treatments involving this reporter, the authors need to do a control where they demonstrate PKD-dependence of the observed increase in reporter signal by including a PKD-knockdown control.

[Editors' note: further revisions were requested prior to acceptance, as described below.]

Thank you for resubmitting your work entitled "A Rho signaling network links microtubules to PKD controlled carrier transport to focal adhesions" for further consideration at *eLife*. Your revised article has been favorably evaluated by Vivek Malhotra as the Senior Editor, Suzanne Pfeffer as the Reviewing Editor, and three reviewers.

We would like to publish your paper but one reviewer wrote,"… in their response to point 2, they claim that they have toned down the message that GEF-H1 mediates activation of RhoA at the Golgi. I do not see a clear indication for this. The reader is still led to believe that GEF-H1 specifically regulates the Golgi pool of RhoA. This is strange because the authors concede that GEF-H1 regulates RhoA in the entire cell, including the Golgi pool." If you can add a few words to make sure this is clarified, we will be delighted to publish your paper in *eLife*.

---

## [Author Response]

This manuscript investigates the RhoA-dependent activation of PKD at the Golgi. While RhoA has been observed at the Golgi over a decade, it has not been clear what this small GTPase is doing at this location. The current work establishes a RhoA-PLCε-PKD signaling cascade that operates at the Golgi and is important for maintaining Golgi structure and for the export of Rab6 carriers from this organelle. The work is important but the following issues should be addressed before publication can be recommended.1) The most important concern is whether the pathway studied is truly physiological. For example, the experiment in Figure 4C, in which PKD activity is assessed under basal conditions (i.e., not treated with nocodazole, trypsin, or GTP-locked RhoA), indicates that PLCε plays only a very minor role in basal PKD activity. Does PAR activation actually alter secretion level (e.g. of TNFα)? If so, then to what extent is the level changed? And, what is a likely physiological context for this regulation? There are many papers showing the impact of altering intracellular signaling factors (including the ones studied here) on secretory pathway organization and/or function. What is missing is a demonstration that these changes occur in a physiologically relevant signaling condition with the expected changes in output (i.e. added ligand X causing a Y-fold change in localized secretion of Z). Here, we believe it would greatly improve the impact of the study to at least quantify PKD-dependent secretion of TNFα in response to activation of a PAR (one wishes something more selective than trypsin could be used for the activation).

Basal and stimulus dependent PKD activity at the Golgi are dependent on the amount of DAG in TGN membranes. There are several pathways, e.g. ceramide transfer from the ER or the PLC-mediated hydrolysis of PI4P or PI4,5P2 that contribute to the DAG level at this organelle. These different sources of DAG provide an explanation for the modest but significant contribution of PLCε to basal PKD activity as shown in Figure 4C.

To use a more selective and physiological stimulus for the PAR receptors, in the revised manuscript, we treated cells expressing the PKD reporter with thrombin. Our data clearly show that thrombin activates PKD at the Golgi complex (new Figure 4—figure supplement 1). Most importantly, using TIRF kymography, we show that thrombin stimulates the localized exocytosis of Rab6 vesicles at FA in a PKD-dependent manner (new Figure 7E). Previous reports describing thrombin-induced, Rab6-dependent stimulation of TNFα secretion in monocytes support the physiological relevance of a pathway that links PAR signaling with GEF-H1-dependent PKD activation at the TGN (see Discussion of the revised manuscript).

2) The FRET experiment in Figure 2D shows that depletion of GEF-H1 results in a reduction in the total cellular activity of RhoA (lower panel). Therefore, GEF-H1 does not seem to be a Golgi-specific regulator of RhoA. In the upper row of the same panel, treatment with nocodazole also increases total cellular RhoA FRET (i.e. activity). It is hard to see the Golgi in this image, most likely because of fragmentation induced my microtubule depolymerization. It is hard to see Golgi-localized FRET signals in the images provided, even zoomed in. Based on this experiment it is not possible to say that GEF-H1 regulates RhoA activity at the Golgi. Rather, GEF-H1 regulates the total cellular pool of active RhoA, including the Golgi pool.

Indeed, the FRET data show that nocodazole treatment promotes Rho activity not only at Golgi membranes but also in the periphery in a GEF-H1 dependent manner. We also mention this in the manuscript. We agree with the reviewers that GEF-H1 seems to regulate different Rho pools and have thus changed the phrasing in the Results section. We have also included this point in the Discussion section.

3) Is GEF-H1 localized at the Golgi? Also, in Figure 1C, the GFP-RhoA-Q63L image shows that this fusion protein localizes to the Golgi in only a subset of the cells. The extent of Golgi localization should be quantified and reported. It appears that the corresponding RhoB construct has much better Golgi localization.

In the new Figure 1—figure supplement 1 we provide quantitative data (Manders coefficient) showing that RhoA and RhoB co-localize with Golgi-localized Rab6 to a similar extent. However, active RhoA and RhoB show significantly higher co-localization compared to their wt counterparts. This also supports a role for active Rho at TGN membranes.

4) The G-PKDrep reporter is heavily utilized in this work, and has been previously validated. However, when new treatments are used that have not previously been validated, it is important to verify that the reporter is indeed reporting PKD activity, as it is easy to imagine that another kinase could phosphorylate the reporter. For example, how do we know that GTP-locked RhoA does not activate another kinase that is able to phosphorylate the G-PKDrep reporter? Or that Trypsin treatment doesn't activate another kinase that will phosphorylate the reporter? Therefore, for any new experimental treatments involving this reporter, the authors need to do a control where they demonstrate PKD-dependence of the observed increase in reporter signal by including a PKD-knockdown control.

To show PKD-specific phosphorylation of the reporter we now provide data showing that pharmacological inhibition of PKD using the pan-PKD inhibitor CRT0066101 completely abrogates the thrombin and trypsin induced phosphorylation of the reporter (new Figure 4—figure supplement 1).

We also show that the depletion of PKD2 and PKD3 significantly decreases trypsin-induced reporter phosphorylation compared to control cells, proving PKD specificity of the reporter (Figure 4F). The greater variance in the signal compared to pharmacological inhibition can be explained by the different transfection efficiency of the siRNAs.

[Editors' note: further revisions were requested prior to acceptance, as described below.]

We would like to publish your paper but one reviewer wrote,"… in their response to point 2, they claim that they have toned down the message that GEF-H1 mediates activation of RhoA at the Golgi. I do not see a clear indication for this. The reader is still led to believe that GEF-H1 specifically regulates the Golgi pool of RhoA. This is strange because the authors concede that GEF-H1 regulates RhoA in the entire cell, including the Golgi pool." If you can add a few words to make sure this is clarified, we will be delighted to publish your paper in eLife.

As requested by the reviewer, we have made changes to the manuscript text especially in the Results section to clarify that GEF-H1 promotes a general increase in cellular Rho activity and that active Rho in turn stimulates PKD activity at the TGN membranes.

For example, we corrected the sentence “Our data thus provide strong support for GEF-H1 promoting increased Rho activity at Golgi membranes.” to “Our data thus provide strong support for GEF-H1 promoting increased Rho activity *in the cell periphery* and at the Golgi membranes.”

We also corrected the sentence “Specifically, we demonstrated that GEF-H1, when released from microtubules, enhances RhoA activity at TGN membranes. Active RhoA in turn induces PKD activation through its effector PLCε ultimately leading to the fission of Rab6-positive vesicles that move towards FAs.” to “Specifically, we demonstrate that GEF-H1, when released from microtubules, enhances cellular RhoA activity, which in turn promotes PKD activation at the TGN through the Rho effector PLCε, ultimately leading to the fission of Rab6-positive vesicles that move towards FAs.”

In total, we changed the phrasing in nine sentences. With these changes the reader is no longer misled to assume that GEF-H1 specifically activates Rho at the Golgi complex.